# Let Your Light Shine: Foreground Portrait Matting via Deep Flash Priors

**Tianyi Xiang**                                    *tianxiang6-c@my.cityu.edu.hk*
*Department of Computer Science, City University of Hong Kong*
*School of Computing and Information Systems, Singapore Management University*

**Yangyang Xu**[†]                                    *xuyangyang@hit.edu.cn*
*School of Intelligence Science and Engineering, Harbin Institute of Technology (Shenzhen)*

**Qingxuan Hu**                                    *qxhbbxx@gmail.com*
*School of Computing, National University of Singapore*
*School of Computing and Information Systems, Singapore Management University*

**Chenyi Zi**                                    *barristanzi666@gmail.com*
*Department of Data Science and Analytics, HKUST (Guangzhou)*
*School of Computing and Information Systems, Singapore Management University*

**Nanxuan Zhao**                                    *nanxuanzhao@gmail.com*
*Adobe Research*

**Junle Wang**                                    *wangjunle@gmail.com*
*Tencent*

**Shengfeng He**[†]                                    *shengfenghe@smu.edu.sg*
*School of Computing and Information Systems, Singapore Management University*

**Reviewed on OpenReview:** *https://openreview.net/forum?id=vxUiVJp2eM*

## Abstract

In this paper, we delve into a new perspective to solve image matting by revealing the foreground with flash priors. Previous Background Matting frameworks require a clean background as input, and although demonstrated powerfully, they are not practical to handle real-world scenarios with dynamic camera or background movement. We introduce the flash/no-flash image pair to portray the foreground object while eliminating the influence of dynamic background. The rationale behind this is that the foreground object is closer to the camera and thus received more light than the background. We propose a cascaded end-to-end network to integrate flash prior knowledge into the alpha matte estimation process. Particularly, a transformer-based Foreground Correlation Module is presented to connect foregrounds exposed in different lightings, which can effectively filter out the perturbation from the dynamic background and also robust to foreground motion. The initial prediction is concatenated with a Boundary Matting Network to polish the details of previous predictions. To supplement the training and evaluation of our flash/no-flash framework, we construct the first flash/no-flash portrait image matting dataset with 3,025 well-annotated alpha mattes. Experimental evaluations show that our proposed model significantly outperforms existing trimap-free matting methods on scenes with dynamic backgrounds. Moreover, we detailedly discuss and analyze the effects of different prior knowledge on static and dynamic backgrounds. In contrast to the restricted scenarios of Background Matting, we demonstrate a

---

[†]Corresponding authors.

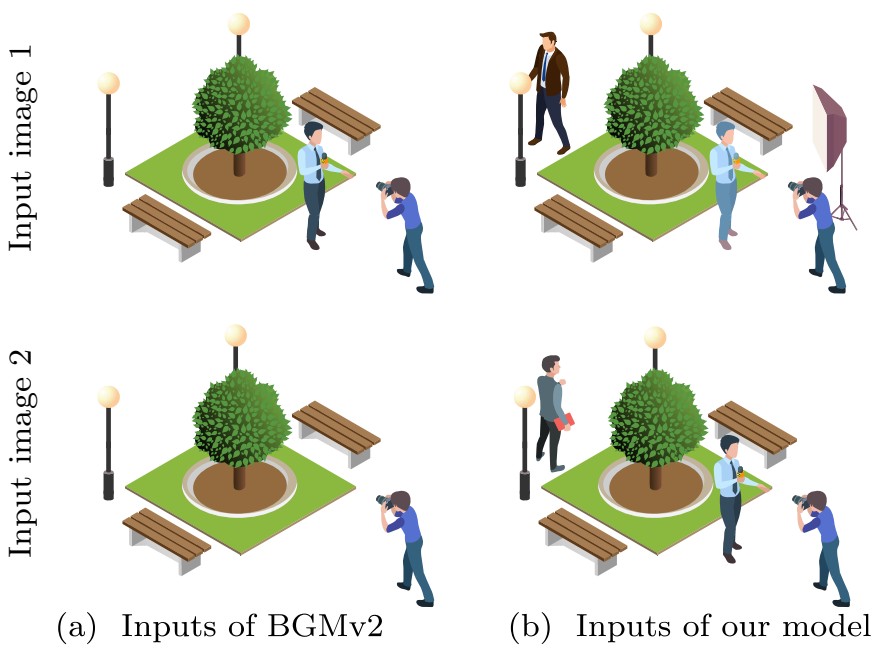

(a)  Inputs of BGMv2          (b)  Inputs of our model

Figure 1: We consider the portrait matting problem in the dynamic scene. BGMv2 (Lin et al., 2021) requires two images (a) as inputs. However, their assumption of a clean and static background image does not hold in a dynamic scene, in which it produces a poor alpha matte. We propose a new setting to use an additional flash image to shed light on the foreground object. Our setting is robust to misalignments like background objects and camera movements.

flexible and reliable solution in real-world cases with the camera or background movements. The code is available at `https://github.com/xty435768/DeepFlashMatting`.

# 1  Introduction

Image matting has been a core and fundamental problem in computer vision that extensively applied in film production, social media, video conferencing, and many other fields. Nowadays, video conferencing has become the status quo in business and educational contexts, and image matting can be used to prevent privacy leakage by replacing a virtual background. Mathematically, the image matting problem can be formulated as

$$I = \alpha F + (1 - \alpha)B, \tag{1}$$

where the foreground $F$, background $B$, and the pixel-wise opacity $\alpha$ are unknowns. Given an input image $I$ in RGB representation, estimating seven unknown factors are obviously an ill-posed problem.

To resolve this problem, traditional matting methods (Ruzon & Tomasi, 2000; Chuang et al., 2001; Wang & Cohen, 2007; Gastal & Oliveira, 2010; He et al., 2011; Feng et al., 2016) usually rely on a trimap as a prior to limit the solution space. However, generating a high-quality trimap is time-consuming and impractical in real-time applications. Recently, some trimap-free methods are proposed to replace the trimap with other prior knowledge. Such of methods adopt the clean background (Sengupta et al., 2020; Lin et al., 2021), semantic mask (Chen et al., 2018; Yu et al., 2021b; Park et al., 2023; Jiang et al., 2025), or temporal features in videos (Sun et al., 2021b; Lin et al., 2022; Li et al., 2023a; 2024a;b; Lin et al., 2023), as auxiliaries to predict the alpha matte.

Among these methods, Background Matting series (*i.e.*, BGMv1 (Sengupta et al., 2020) and BGMv2 (Lin et al., 2021)) achieves robust performances by taking a clean background as an additional input (see Fig. 1a). This clean and static background provides a very strong prior knowledge that objects that have not appeared

in the background image must belong to the foreground. With great assumption comes great restriction, background matting cannot handle any types of movements that make the source image not aligned with the background image.

In this paper, we resolve this problem from an opposite perspective. Instead of providing background hints, we shed light on the foreground regions with flash photography. The rationale behind is that foreground objects are typically closer to the camera and thus received more lights than the background. Taking an additional flash input can not only provide strong foreground clues, it also enables a more flexible and practical setting: 1) users are not required to take the additional reference image in advance (*e.g.*, background image in BGM), and the matting process can start together with filming; 2) main objects can move around, and we can take a new flash image anytime during filming to adapt to the new scene.

To fully exploit flash priors, we devise the Flash and No-flash Net (FNFNet), which uses flash/no-flash image pair to produce high-resolution portrait alpha matte and the foreground of the no-flash image. The estimation is insensitive to moving foreground, background misalignment, and camera shaking. Our FNFNet consists of two cascaded stages. The first stage aims to explore the interaction between flash and no-flash images, and provides coarse estimations of trimap and initial alpha matte. Concretely, we propose a transformer-based Foreground Correlation Module (FCM) to jointly identify illumination-irrelevant foreground appearances across two inputs in a reciprocal manner. Moreover, we calculate a flash ratio map between flash/no-flash images as a coarse indicator of foreground regions to supplement the alpha matte estimation. In the second stage, the trimap and initial alpha matte are fed to the Boundary Matting Network, which has a unique emphasis on producing fine boundary details of the mattes.

In order to train and evaluate our method and promote research on flash photography, we construct the first flash/no-flash portrait matting dataset. It consists of more than 100 diverse videos captured using the green screen, in total containing 3,025 well-annotated alpha mattes. We also collect dynamic videos of various real-world scenes to compose dynamic background image pairs. Therefore our benchmark can be used to evaluate matting methods with static and dynamic backgrounds. Several trimap-free matting methods are compared on this benchmark. Extensive experiments show that our proposed method achieves superior performances on the scenarios with dynamic backgrounds, and our flash prior demonstrates comparable performance to the background prior in static scenes.

In summary, our contribution is fourfold:

- We construct the first flash/no-flash portrait matting dataset. It consists of more than 100 diverse videos with 3,025 well-annotated alpha mattes. We collect additional dynamic background videos, together with static backgrounds, to form a benchmark for evaluating the matting performance on both scenarios.

- Rather than relying on a clean background, we propose a deep flash prior, which enables a more flexible and practical solution for real-time or live-streaming matting.

- We propose FNFNet to integrate flash photography into a deep neural network. It explores the interaction between two differently illuminated foregrounds to extract high-quality alpha matte.

- Experiment results show that our model achieves state-of-the-art performance among other trimap-free matting methods. More importantly, our method is insensitive to the dynamic background and not restricted to well-aligned foreground or background between source and reference images.

## 2  Related work

### 2.1  Trimap-based Matting

Traditional methods often require a manually annotated trimap as additional information for solving the matting problem, which can be classified into sampling-based methods and propagation-based methods. Sampling-based methods (Ruzon & Tomasi, 2000; Chuang et al., 2001; Wang & Cohen, 2007; Gastal & Oliveira, 2010; He et al., 2011; Feng et al., 2016; Johnson et al., 2016) sample pixels from the labelled

foreground and background regions to get the color information, and then estimate the alpha matte of the unknown region. Propagation-based methods (Sun et al., 2004; Levin et al., 2007; 2008; Chen et al., 2013; Aksoy et al., 2017) build the similarity of neighboring pixels, trying to propagate alpha value from known to unknown regions. Since the above methods only utilize low-level features, they seldom achieve good performance in complex scenes. With the rapid development of deep learning, many methods (Yao et al., 2024; Cai et al., 2022; Park et al., 2022; Lutz et al., 2018; Tang et al., 2019; Liu et al., 2021c) based on neural network have been explored and have made unprecedented progress in image matting, which have been comprehensively reviewed in Li et al. (2023b); Lepcha et al. (2023). Xu *et al.* (Xu et al., 2017) proposes a two-stage neural network structure named Deep Image Matting (DIM) to obtain the coarse alpha and refined alpha matte, respectively. They also collects the first large-scale matting dataset, which empowers the development of many following works. Hou *et al.* (Hou & Liu, 2019) employs two encoders to extract local features and the global context information respectively, and estimates foreground and alpha simultaneously. Lu *et al.* (Lu et al., 2019) presents IndexNet, which can dynamically generates indices as conditions to guide the matting context modelling and operating. GCAMatting (Li & Lu, 2020) designs a Guided Contextual Attention module to propagate global high-level features based on the learned low-level affinity. Cai *et al.* (Cai et al., 2019) disentangles image matting into two sub-tasks: trimap adaptation and alpha estimation, with a disentangled framework proposed named AdaMatting to solve them. Yu *et al.* (Yu et al., 2021a) extend the matting problem to high-resolution images relying on a patch-based method with several modules to capture long-range matting contextual and handle information propagation. Zhang *et al.* (Zhang et al., 2021) suggest propagating the trimap via temporal dimension to achieve video object matting using a temporal aggregation module. In Sun et al. (2021a) and Liu et al. (2021a), semantic classification of matting regions is incorporated into the matting framework. Recently, some methods leverage diffusion algorithm (Ho et al., 2020) to facilitate image matting, including diffusing in pixel space by diffusing a disturbed trimap (Xu et al., 2023) or pure noise (Hu et al., 2024) until a clean alpha matte is produced, or diffusing in latent space (Li et al., 2024d). However, as trimap-based methods rely on a well-annotated trimap as the input for generating plausible results, they often fail to be applied in real-time scenarios.

## 2.2 Trimap-free Matting

Many recent works (Zhang et al., 2019; Dai et al., 2022; Zhu et al., 2017; Shen et al., 2016; Chen et al., 2018; Liu et al., 2020; Yu et al., 2021b; Qiao et al., 2020; Park et al., 2023; Jiang et al., 2025) try to relax trimap input by adding a segmentation process before acquiring the final alpha matte, while matting performance overly depends on the results of segmentation. HAttMatting (Qiao et al., 2020) introduces a blended attention mechanism to integrate features. Self-guidance provided by Progressive Refinement Network (PRN) module in Yu et al. (2021b) can deal with versatile guidance masks. LFPNet (Liu et al., 2021b) learns long-range context features outside the reception field and propagates learned surrounding features to help matting prediction inside the reception field. Sun *et al.* (Sun et al., 2022) introduce flaw detection as a sub-task to achieve automatic error-correcting in trimap-free portrait matting problems. Lin et al. (2022); Li et al. (2023a); Sun et al. (2023); Li et al. (2024a); Gu et al. (2023); Li et al. (2024b); Lin et al. (2023) adopt temporal information in the video as guidance to do the video objects or portrait matting. However, the temporal priors relied on by these video matting methods may face challenges in various dynamic backgrounds. Recently, some trimap-free matting methods (Ke et al., 2022; Chen et al., 2022b;a; Li et al., 2022; Ma et al., 2023) follow the idea of decomposing the matting problem into semantic prediction and detail prediction and then considering fusion or collaboration between these two sub-objectives. However, these methods usually perform worse than trimap-based methods in matting quality. Some methods take advantage of the text-image correspondence prior (*e.g.*, CLIP (Radford et al., 2021)) to achieve referring image matting (Xiang et al., 2025; Li et al., 2023c). Some methods also try to directly solve the alpha via in-context prior (Guo et al., 2024), latent diffusion prior (Wang et al., 2024), or vision foundation model (Ye et al., 2024; Li et al., 2024c). Sengupta *et al.* (Sengupta et al., 2020) proposes Background Matting (BGM) which only requires a pre-captured background image without foreground objects as auxiliary cues to help determine the position of the foreground. This method achieves high-quality results. Based on Sengupta et al. (2020), Lin *et al.* (Lin et al., 2021) further uses a two-stage network to realize real-time high-resolution matting. However, background matting cannot handle dynamic backgrounds and large camera movements.

## 2.3 Flash Photography Applications

The advantages of exploiting flash photography have been explored in many computer vision tasks. For adopting flash and no-flash image pairs, specially, Sun *et al.* (Sun et al., 2006) tries to recover the matte based on the simple observation that the most noticeable difference between the flash and no-flash images is the foreground object if the background scene is far enough away. Additional information given by flash/no-flash image pairs is also explored in He & Lau (2014) for saliency detection. However, they strictly require perfect alignment for the foregrounds in flash and no-flash images, making it unsuitable for scenarios involving dynamic foregrounds. Sun *et al.* (Sun et al., 2007) suggests adopting a Markov Random Field to combine flash, motion, and color cues and model the flash effect using the histograms of the flash/no-flash images. However, it strongly relies on several strict low-level feature assumptions like the background consistency (*i.e.*, near-bijective relationship for the background region) and obvious contrast due to flash in foreground regions in flash/no-flash image pairs, making it fail when the foreground undergoes significant pose variations, or when the background exhibits dynamic changes (*e.g.*, new objects appearing), or the flash contrast for the foreground is not obvious.

In our work, we explore the previously unaddressed connection between flash cues and high-level instance-level representations. By modeling instance-level matching representations, our method effectively tackles the challenges of dynamic flash matting, including variations in foreground pose and position, as well as interference from dynamic backgrounds.

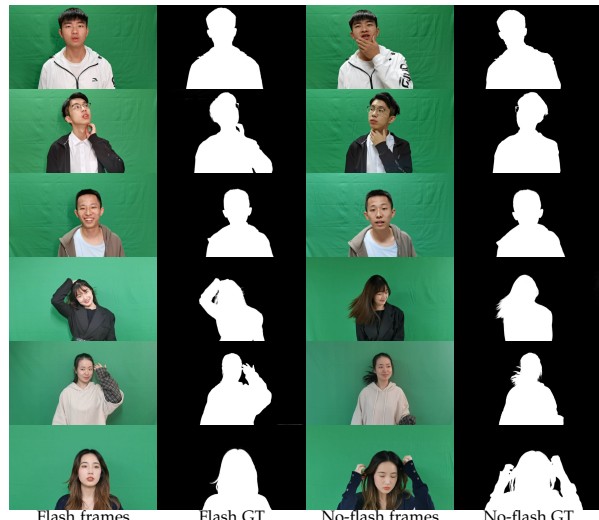

Flash frames     Flash GT     No-flash frames     No-flash GT

(a) Examples of selected flash/no-flash portrait frames and corresponding GT alpha matte in our dataset.

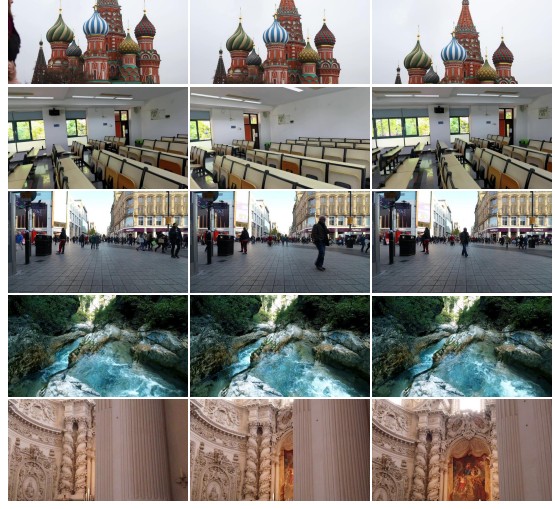

(b) Examples of dynamic backgrounds in our dataset.

Figure 2: Examples from our dataset: (a) flash/no-flash portrait frames with ground truth alpha mattes, and (b) dynamic backgrounds.

# 3 FNF Matting Dataset

To the best of our knowledge, there does not exist an available large-scale dataset containing flash and no-flash image pairs that can be used for portrait matting. Therefore, we construct the first flash and no-flash portrait matting dataset by combing foreground portraits (Fig. 2a) and dynamic background scenes (Fig. 2b) together.

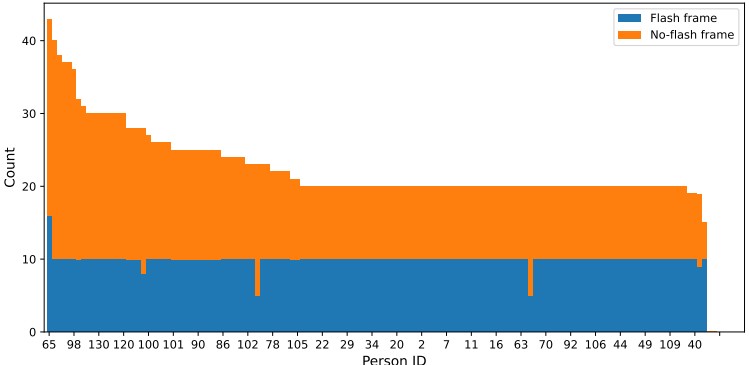

(a) The number of flash/no-flash frames per person, sorted in decreasing order of the total number of frames selected for each person.

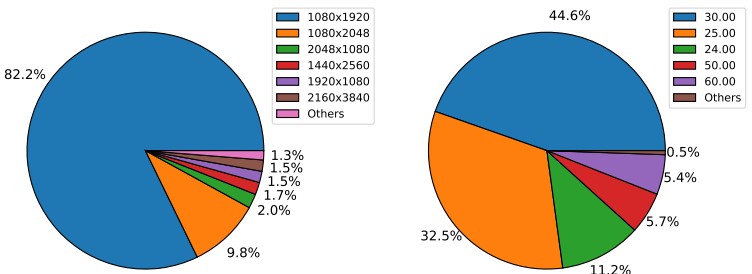

(b) The distributions of resolution (left, height×width) and frame rate (right, fps) of dynamic backgrounds.

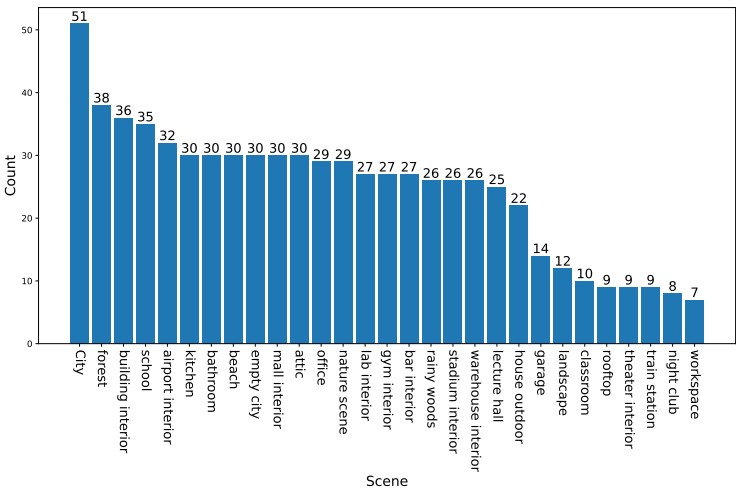

(c) The number of background videos under different scenes.

Figure 3: The statistics of the proposed FNF Matting dataset. We show that the dataset has a rational distribution of flash/no-flash foregrounds for each green-screen portrait video. We also illustrate some basic properties (resolution and frame rate) of collected dynamic background videos and distribution in various scenes.

## 3.1 Data Collection and Processing

**Foreground Portrait.** To collect the foreground portrait videos, we take the green screen as the background and use the two latest smartphones for shooting. We invited 133 participants (66 males and 67 females) for taking the videos. Each video is in a resolution of 1080P and lasts around 20 seconds. Participants

Table 1: Statistics of our proposed dataset.

|  | Training Set | Test Set | Total |
|---|---|---|---|
| Protrait Videos | 111 | 22 | 133 |
| Flash Frames | 1,102 | 220 | 1,322 |
| No-flash Frames | 1,408 | 295 | 1,703 |
| Dynamic Background Videos | 625 | 89 | 714 |
| Dynamic Background Frames | 11,806 | 1,712 | 13,518 |

were asked to act on some common actions that happened during the online meeting for simulating the real scenarios, such as turning around, talking, and playing with their hair. A hand-held lamp is used for generating the flash and no-flash effects. We ensure the lighting conditions and camera settings are consistent across all videos, to minimize individual bias and improve annotation convenience. For each video, we use the Super Key function in Adobe Premiere to extract the foregrounds and alpha mattes automatically. In practical, process a single video only takes less than 10 minutes, and the entire annotation process for the dataset was completed in several days with 3 skilled workers. As shown in Fig. 2a, our dataset covers diverse foreground portraits under common matting cases, such as long and spread hair (*e.g.*, 4th row of Fig. 2a) and transparent area (*e.g.*, eyeglasses in 2nd row of Fig. 2a). The portraits are also in different clothes and poses to enhance the robustness of the model. In total, we obtain 1,322 flash and 1,703 no-flash portraits with their well-annotated alpha mattes.

**Dynamic Background.** For background, we collect 714 videos under diverse real-world scenes from the Internet. We sample frames from each video with an interval of 15, and we only select 20 frames at most from each video. Several examples have been shown in Fig. 2b.

We split the dataset for training and testing in the video-level and show the statistics in Tab. 1.

### 3.2   Dataset Statistics Analysis

To gain more knowledge from our dataset, we conduct some statistic analysis in this subsection and show the results in Fig. 3.

#### 3.2.1   Flash/no-flash Frames

The number of selected flash/no-flash frames of each person (*i.e.*, video) is shown in Fig. 3a. We can see that majority of video has no less than 20 flash and no-flash frames in total. Some video has more than 20 selected frames, indicating they contain more various poses or complexity in the edge area of the portrait (*e.g.*, hair, eyeglasses, etc.) relatively. Videos with 20 selected frames are common in portrait variety and normal quality, while videos with poor recording quality have less than 20 selected frames.

#### 3.2.2   Resolution and Frame Rate of Dynamic Backgrounds

The left chart of Fig. 3b shows the resolution distribution of dynamic background videos. It can be seen that more than 80 percent of the videos have a resolution of $1080 \times 1920$, and almost all videos are high resolution. And the vast majority of videos are horizontal, and a few videos are vertical. The right chart of Fig. 3b shows the frame rate (fps) distribution of our dataset. In this chart, we approximate some videos with standard frame rates to the nearest integer frame rates when counting statistics (*e.g.*, video in 29.97 fps is counted as 30 fps). Nearly half of the videos have a frame rate of 30 fps, and most of the videos are recorded at a common frame speed. About 10 percent of the videos have a higher frame rate (*e.g.*, 50 fps, 60 fps, etc.), which means that the difference between two adjacent frames of these videos may be smaller in general.

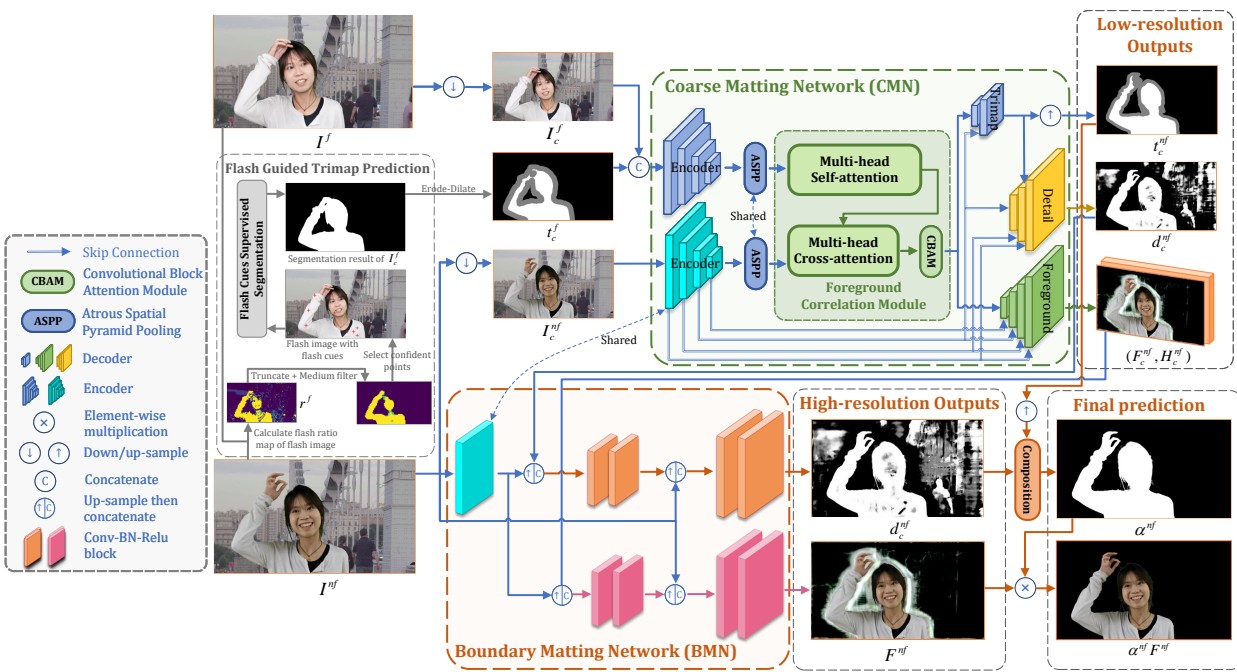

Figure 4: Overview of our FNFNet structure. Given a pair of flash/no-flash images $\{I^f, I^{nf}\}$ as input, we first down-sample them with a scale factor $c'(c' \leq 0.1)$ to $\{I^f_{c'}, I^{nf}_{c'}\}$, and calculate the smoothed flash ratio map $(r^f_p)$ of the flash image according to the interior transfer of pixels affected by flash in the image histograms. Then we randomly select $q$ points from the truncated ratio map as flash cues, and adopt a pretrained segmentation model to extract the trimap $t^f_c$ of the flash image $I^f_c$ with down-sample scale $c(c \leq 0.5)$. The Coarse Matting Network (CMN) takes $\{I^f_c, I^{nf}_c, t^f_c\}$ as inputs and predicts low-resolution matting results of the no-flash image: trimap $t^{nf}_c$, detail map $d^{nf}_c$, foreground color $F^{nf}_c$ with context hidden feature $F^{nf}_c$. The Boundary Matting Network (BMN) is further introduced to enhance the fine details and up-sample $d^{nf}_c$ and $F^{nf}_c$ to the original resolution $(d^{nf}, F^{nf})$.

### 3.2.3 Scenes of Background Videos

Our dataset consists of background videos across 29 categories, and the specific statistics are shown in Fig. 3c. As can be seen, the category "city" is the most common one in our dataset, as most of our background videos come from urban street views. There are also many videos shot in places such as building interiors and natural landscapes. Almost all of them record dynamic changing due to camera shaking or movement, pedestrian or vehicle movement, water flowing, or other dynamic changes. Examples of such changes can also be found in Fig. 2b.

### 3.3 Mitigating Synthetic Data Discrepancy

In light of the disparities between composite data used for training and natural data, such as differences in resolution, noise distribution, and composite artifacts between the foreground and background, we have adopted techniques from Li et al. (2022) to address these discrepancies. To tackle resolution differences, we have meticulously selected background videos with high resolutions that align with those of the foreground videos, as elaborated in Sec. 3.2.2. In addressing differences in noise distribution, we initiate the process by applying BM3D (Dabov et al., 2009) for denoising both the foreground and background. Subsequently, we introduce Gaussian noise with the same deviation to further refine the composition. To further mitigate composite artifacts, we incorporate the re-JPEG (Hou & Liu, 2019) operation into our data augmentation strategy for training samples.

# 4 Method

Considering the person in front of the camera cannot remain completely still and the background may also undergo some dynamic changes in real-world video conference scenes, we need to take full advantage of the assumption that flash has a significant enhancement of foreground color intensity. This leads us to establish a non-local relationship between foreground person in flash and no-flash images, providing foreground location cues to guide the matting of no-flash image.

To this end, we propose the Foreground Correlation Module (FCM), a dual-branch and Transformer-based component to integrate implicit foreground information extracted from flash image to no-flash image features. Moreover, inspired by Sun et al. (2007); He & Lau (2014), we introduce the flash ratio map as another flash cue, imprecisely labelling pixels in both flash and no-flash images with significant intensity changes as foreground based on the image histogram. We furthermore embed FCM to the Flash and No-flash matting network (FNFNet) to obtain robust and high-quality alpha matte of the no-flash image in the dynamic background scene.

## 4.1 Model Overview

Fig. 4 demonstrates the pipeline of our proposed FNFNet. FNFNet includes a Coarse Matting Network (CMN) and a Boundary Matting Network (BMN). The CMN has two inputs, one is the down-sampled no-flash image $I_c^{nf}$, the other is the concatenation of flash image $I_c^f$ with its trimap $t_c^f$ (which is obtained by flash cues guided trimap prediction), two inputs are extracted to their high-level features respectively. Then these two features are fed into the Foreground Correlation Module (FCM) for feature aggregation, following three separate decoders to predict the trimap $t_c^{nf}$, detail map $d_c^{nf}$, and foreground prediction $F_c^{nf}$ of the no-flash image. In BMN, we take the original no-flash image and reuse the first three layers of the encoder in CMN to extract low-level textual features, enabling BMN to recover details and up-sample the low-resolution outputs from CMN to produce the high-resolution detail map $d^{nf}$ and foreground prediction $F^{nf}$ of the no-flash image. Then the alpha matte $\alpha^{nf}$ of the no-flash image can be obtained by compositing the up-sampled trimap $t^{nf}$ and detail map $d^{nf}$.

## 4.2 Flash Guided Flash-Trimap Prediction

The additional flash provides strong cues on revealing foreground objects. Before getting into the network, we first derive an extra input of our framework, the trimap of flash image predicted using flash ratio map.

### 4.2.1 Flash ratio map

The flash ratio map is obtained by analyzing the differences between flash and no-flash histograms. The reason for utilizing the image histogram is that it gives the distribution of the color intensity of each pixel from the overall statistical level, making the produced flash ratio map insensitive to small background changes and foreground movements.

We denote the RGB color histogram of the flash images as $H^f = \{h_k^f\}$, and no-flash image histogram as $H^{nf} = \{h_k^{nf}\}$, where $h_k$ indicates the corresponding number of pixel in the $k$-th bin. Here, both $H^f$ and $H^{nf}$ are computed in the entire RGB space, which ensures that the same index value (*i.e.*, "global" index) refers to the same color bin in both histograms, ensuring consistency across both images. We observe that the flash will modify and transfer some pixels in the low-intensity bins of the no-flash image histogram to the high-intensity bins of the flash image histogram (illustrated in Fig. 5), we have the following analysis with $h_k^f$ and $h_k^{nf}$.

For $h_k^f > h_k^{nf}$, we can infer that some pixels in low-intensity bins of no-flash histogram have been transferred to the $k$-th bin of flash image histogram due to the flash effect. Thus, pixels in the $k$-th bin of the flash image histogram are more likely to be the foreground. While $h_k^f < h_k^{nf}$ indicates that some pixels in the $k$-th bin of the no-flash image histogram have been transferred to high-intensity bins of the flash image histogram. Similarly, pixels in the $k$-th bin of the flash image histogram have a higher probability of being the foreground.

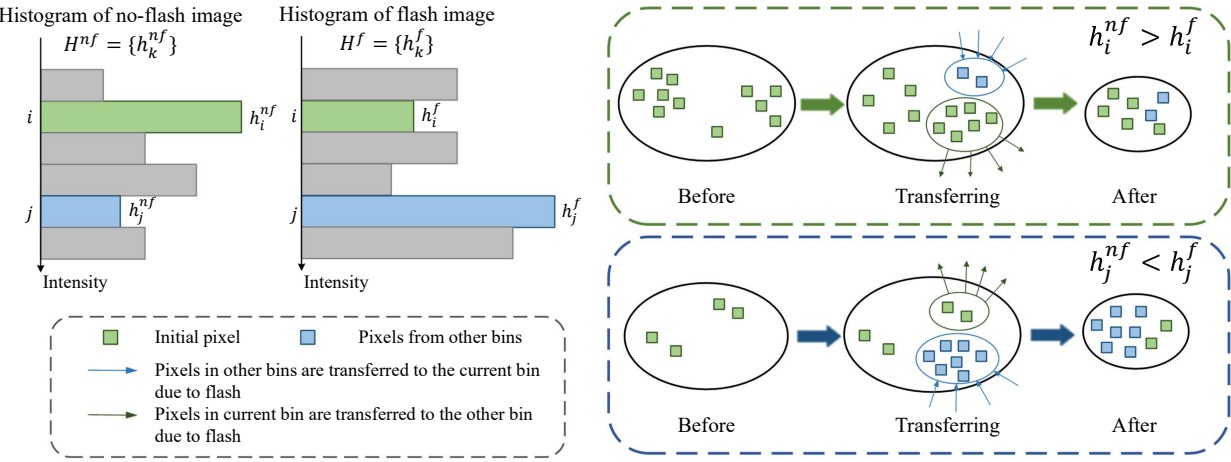

Figure 5: Demonstration of our flash ratio map. Foreground pixels are normally transferred from low intensity to high intensity due to the flash. Takes the $i$-th bin in the histograms of no-flash and flash image ($h_i^{nf}$ and $h_i^f$ respectively) as an example. If $h_i^{nf} > h_i^f$, it means that the number of pixels transferred to other bins is more than the newly added pixels. Therefore, pixels in the $i$-th bin of the no-flash image are more likely to be foreground. Similar conclusion can be found in the $j$-th bin of the flash image if $h_j^{nf} < h_j^f$.

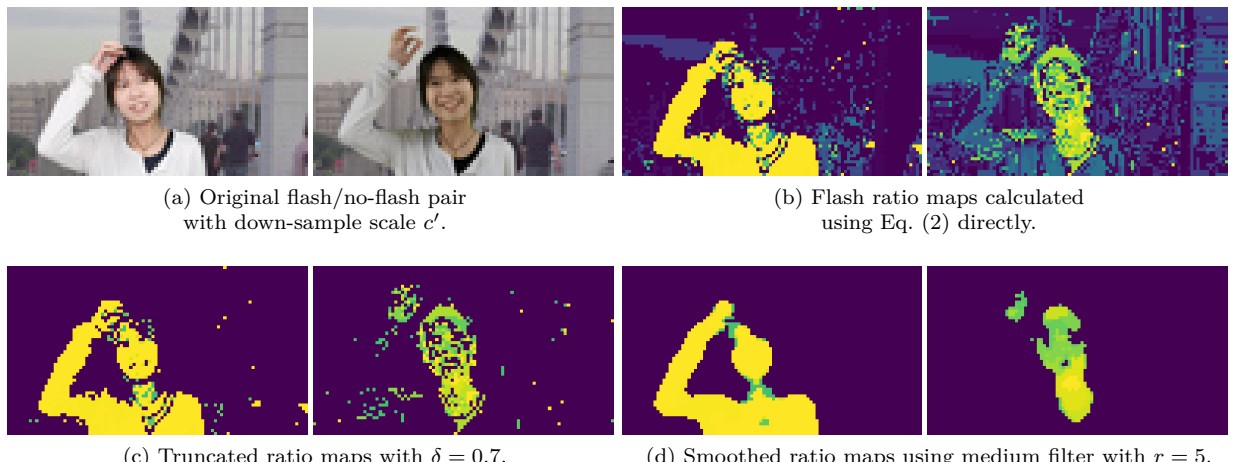

(a) Original flash/no-flash pair with down-sample scale $c'$.

(b) Flash ratio maps calculated using Eq. (2) directly.

(c) Truncated ratio maps with $\delta = 0.7$.

(d) Smoothed ratio maps using medium filter with $r = 5$.

Figure 6: Examples of flash ratio maps. For pairs of down-sampled flash/no-flash images in (a), (b) shows the flash ratio maps computed by the differences between histogram distributions. We furthermore refine (b) by increasing truncate threshold $\delta$ (c) and applying medium filtering to get a smoother result (d).

Based on the above analysis, given a pair of flash/no-flash image, their corresponding flash ratio maps $r_p^f$ and $r_p^{nf}$ at pixel $p$ can be quantified as:

$$r_p^f = \frac{h_{k_p}^f - h_{k_p}^{nf}}{h_{k_p}^f}, \quad r_p^{nf} = \frac{h_{k_p}^{nf} - h_{k_p}^f}{h_{k_p}^{nf}}, \tag{2}$$

where $k_p$ is the bin index which contains $p$.

### 4.2.2 Flash-Trimap Processing

In order to get a flash ratio map that can provide a more accurate flash hint for further trimap prediction, we extract the confident points as flash cues by following steps. We first calculate the flash ratio map $r^f$ of the down-sampled flash image $I_{c'}^f$ with the scale factor $\frac{1}{c'}$ using Eq. (2). Particularly, $c'$ is usually set to be less than 0.1 to eliminate interference from irrelevant pixels as much as possible. Here we do not calculate $r^{nf}$ because $r^{nf}$ is often inaccurate while $r^f$ has a more accurate estimate of the foreground in the flash image. In order to retain more reliable regions in $r^f$, we first perform truncation on $r^f$ using following equation:

$$r^f := \begin{cases} r^f, & r^f \geq \delta \\ 0, & otherwise \end{cases} \tag{3}$$

where $\delta$ is a threshold to control the truncation level considering the flash ratio map may not be completely correct. Note that Eq. (3) can also help to eliminate the undefined cases when $h_{k_p}^f$ or $h_{k_p}^{nf}$ is 0 in Eq. (2), since $r$ will become $-\infty$ but can be truncated to 0 directly given positive $\delta$. $r^f$ is further smoothed by applying the medium filter with kernel size $r$ for further refinement.

We showcase intermediate results before and after smoothing in Fig. 6. To ensure that selected points are distributed evenly and do not accumulate in local areas, we randomly choose $q$ points from $r^f$ as flash cues and then map these points to the resolution of $I_c^f$. We employ a pretrained segmentation model to extract the segmentation mask of the foreground, using $I_c^f$ and the flash cues as input. Subsequently, we derive the trimap $t_c^f$ for $I_c^f$ from the segmentation mask using an erosion-dilation operation. During training, to enhance the training process for the subsequent end-to-end network, we directly apply the erosion-dilation operation to the ground truth alpha $\alpha_c^f$ of $I_c^f$ to obtain $t_c^f$.

## 4.3 Coarse Matting Network

Inspired by DeeplabV3 (Chen et al., 2017), our CMN has the encoder-decoder architecture, which includes two separate encoders with ResNet-50 (He et al., 2016) structures, the Atrous Spatial Pyramid Pooling (ASPP) modules, a Foreground Correlation Module (FCM), and three decoder networks for different types of output.

### 4.3.1 Backbone Network

Given a pair of flash and no-flash images $I^f, I^{nf} \in \mathbb{R}^{H \times W \times 3}$, we first down-sample them with a scale factor $\frac{1}{c}$ to $I_c^f, I_c^{nf} \in \mathbb{R}^{\frac{H}{c} \times \frac{W}{c} \times 3}$, and obtain one-hot trimap $t_c^f \in \{0,1\}^{\frac{H}{c} \times \frac{W}{c} \times 3}$ of $I_c^f$. As discussed in Sec. 4.1, $I_c^{nf}$, and the concatenation between $I_c^f$ and $t_c^f$ are fed into the two non-shared encoders following the ASPP module separately to extract their respective feature maps, denoted as $x^f, x^{nf} \in \mathbb{R}^{h \times w \times C}$ respectively. Here, the ASPP module consists of three dilated convolution layers with dilation rates of 3, 6, and 9 to summarize the contextual information.

### 4.3.2 Foreground Correlation Module

As depicted in Fig. 7, we feed $x^f$ and $x^{nf}$ into the FCM. Our objective is to establish an association between the flash image and the no-flash information using a cross-attention mechanism. The leftmost branch, $T^{nf}$, comprises $L_{nf}$ identical cross-attention layers, which are designed to facilitate the connection and fusion of the no-flash information with the foreground in the flash image. On the other hand, the rightmost branch, $T^f$, consists of $L_f$ identical self-attention layers, focusing solely on extracting features from the flash image. Both of these branches take input sequences with a length of $h \times w$. To create these sequences, we apply a linear projection followed by a flatten operation to the input features $x^f, x^{nf}$, resulting in corresponding input sequences denoted as $Z^f, Z^{nf} \in \mathbb{R}^{hw \times C'}$, where $C'$ represents the dimension of the embedding space. Additionally, we incorporate learnable positional embeddings into these sequences.

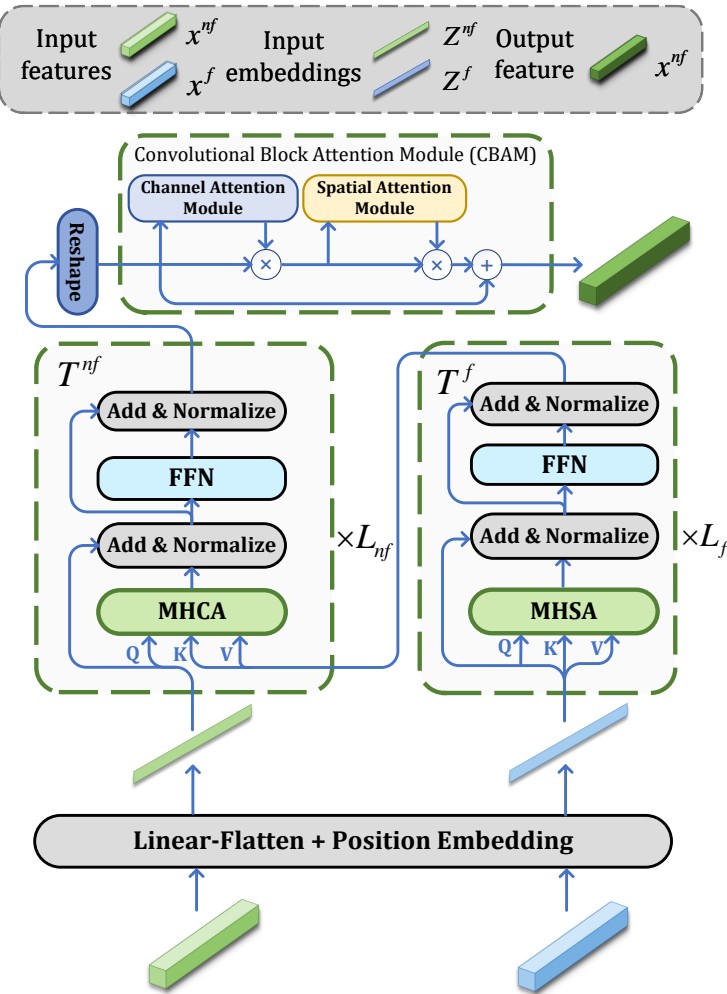

Figure 7: Overview of the Foreground Correlation Module (FCM). FCM consists of two branches: $T^f$ and $T^{nf}$, each of which contains $L_f$ and $L_{nf}$ transformer layers respectively. Each layer in $T^f$ only have Multi-head Self-Attention (MHSA), while in $T^{nf}$, each layer have the Multi-head Cross-Attention (MHCA). The Convolutional Block Attention Module (CBAM) (Woo et al., 2018) is applied for the reshaped feature outputted by $T^{nf}$ for feature re-attention.

**Computation of $\mathbf{T^f}$:** The embedded sequences $Z^f$ is fed into $T^f$. At each layer $l^f$ in $T^f$, we first calculate the triplet $(Q_{l-1}^f, K_{l-1}^f, V_{l-1}^f)$ of $Z_{l-1}^f$, the input of current layer. Then the self-attention $SA_{l-1}^f$ is given as:

$$SA_{l-1}^f = \texttt{softmax}(\frac{Q_{l-1}^f(K_{l-1}^f)^T)}{\sqrt{d}})(V_{l-1}^f). \tag{4}$$

We further calculate the multi-head self-attention $MSA_{l-1}^f$ with the residual connection by concatenating $m$ independent self-attention operation following a linear projection. Then, a feed-forward network (FFN) with the residual connection is used to transform $MSA_{l-1}^f$ and obtain the output $Z_l^f \in \mathbb{R}^{hw \times C'}$ of current layer:

$$Z_l^f = FFN_{l-1}^f(MSA_{l-1}^f) + MSA_{l-1}^f. \tag{5}$$

We denote $Z_{L_f}^f$ as the output of the last layer of $T^f$. Recall that $L_f$ represents the layer number of $T_f$.

**Computation of $\mathbf{T^{nf}}$:** At each layer $l^{nf}$ in $T^{nf}$, we utilize the multi-head cross-attention to fuse flash image features extracted from $T^f$ with the input $Z_{l-1}^{nf}$. For multi-head cross-attention, denoted as $MCA_{l-1}^{nf}$, we

first calculate the query$(Q_{l-1}^{nf})$ from $MSA_{l-1}^{nf}$, and the key $(K_{l-1}^{nf})$ and value$(V_{l-1}^{nf})$ from $Z_{L_f}^{f}$. Then $MCA_{l-1}^{nf}$ can be similarly carried out based on Eq. (4). An FFN with the residual connection is applied for feature refinement to produce the output sequence $Z_{l-1}^{nf}$ of current layer:

$$Z_{l-1}^{nf} = FFN_{l-1}^{nf}(MCA_{l-1}^{nf}) + MCA_{l-1}^{nf}. \tag{6}$$

We denote $Z_{L_{nf}}^{nf}$ as the output of the last layer of $T^{nf}$. Finally, $Z_{L_{nf}}^{nf} \in \mathbb{R}^{hw \times C'}$ is reshaped to shape of $\mathbb{R}^{h \times w \times C}$ and passed to the Convolutional Block Attention Module (CBAM) (Woo et al., 2018) for feature re-attention both in spatial and channel-wise and obtain $\widetilde{x}^{nf}$ for further decoding.

### 4.3.3 Decoders

Considering the distinct appearance of semantic and detail features, we adopt three non-shared decoders $(D_{trimap}, D_{detail}, D_{foreground})$ to predict the low-resolution trimap, detail of alpha prediction, and foreground color. Each of these decoders includes different number of Conv-BN-ReLU groups. In each group, the input feature is bilinear-upsampled with a scale factor 2, and then concatenated with the shortcut feature following a $3 \times 3$ convolution, batch normalization (Ioffe & Szegedy, 2015), and ReLU activation (Nair & Hinton, 2010).

The trimap decoder $D_{trimap}$ only contains 2 groups of Conv-BN-ReLU considering $D_{trimap}$ only responsible for coarse trimap estimation. It takes $\widetilde{x}^{nf}$ as input and outputs the classification probabilities with 3 classes (foreground, background, and unknown region) of all pixel of $I_{4c}^{nf}$ at $\frac{1}{4}$ resolution of $I_{c}^{nf}$, denoted as $s_{4c}^{nf} \in [0,1]^{\frac{H}{4c} \times \frac{W}{4c} \times 3}$. Then we directly up-sample $s_{4c}^{nf}$ to original resolution of $I_{c}^{nf}$, denoted as $s_{c}^{nf} \in [0,1]^{\frac{H}{c} \times \frac{W}{c} \times 3}$. By taking the Argmax operation, $s_{c}^{nf}$ can be easily converted to the trimap prediction, denoted as $t_{c}^{nf} \in \{0,1,2\}^{\frac{H}{c} \times \frac{W}{c}}$.

The detail decoder $D_{detail}$ is responsible for predicting fine alpha matte in the unknown region. For preserving rich low-level contextual features, $D_{detail}$ consists of 3 groups of Conv-BN-ReLU, and takes shortcut feature of no-flash image encoder at $1/8$ resolution of $I_{c}^{nf}$ as input. It also receives the Argmax operation applied $s_{4c}^{nf}$ from $D_{trimap}$ and concatenate it with midden features before going into the second group, fusing semantic information in to better estimate detail alpha. $D_{detail}$ finally outputs the detail alpha matte $d_{c}^{nf} \in [0,1]^{\frac{H}{c} \times \frac{W}{c} \times 1}$, which is only meaningful in the unknown region.

The foreground decoder $D_{foreground}$ consists of 4 groups of Conv-BN-ReLU and also takes $\widetilde{x}^{nf}$ as input. Following Lin et al. (2021), $D_{foreground}$ predicts the foreground residual $R_{c}^{nf}$ and we then add the original image with clamping into $[0,1]$ to obtain the foreground color prediction of the no-flash image, denoted as $F_{c}^{nf} \in \mathbb{R}^{\frac{H}{c} \times \frac{W}{c} \times 3}$. $D_{foreground}$ also produces the hidden features $H_{c}^{nf} \in \mathbb{R}^{\frac{H}{c} \times \frac{W}{c} \times 32}$, which contains the contextual information of foreground. In addition, both $D_{detail}$ and $D_{foreground}$ have skip connections with the no-flash image encoder (see in Fig. 4).

### 4.4 Boundary Matting Network

To obtain the matting result of the no-flash image with the original resolution, we design the boundary matting network to improve the matting result boundaries to high-resolution by incorporating low-level features. For the trimap $t_{c}^{nf}$, we directly take a bi-linear interpolation operation to up-sample it to the original resolution, denoted as $t^{nf}$. And for the detail map $d_{c}^{nf}$ and foreground residual $R_{c}^{nf}$, we treat them differently for enhancement. We first adopt a bi-linear interpolation to up-sample the detail map feature (or foreground feature) to $\frac{1}{2}$ of the original resolution. Then, we reuse the first three layers from the backbone network in CMN, including a $7 \times 7$ convolutional layers, a batch-normalization layer, and a ReLU operation, to extract the low-level feature. We also apply several shallow convolutional layers with a small number of channels to process detailed low-level information. In detail, the low-level feature is concatenated with the detail map feature and passed into two groups of 3x3Conv-BN-ReLU to help restore the matting details. Furthermore, the output feature that has been interpolated to the original resolution, is concatenated with the original no-flash image and passed to another two groups of 3x3Conv-BN-ReLU (with the last ReLU removed) to obtain the final enhanced and high-resolution detail map (or foreground residual), denoted as $d^{nf}$ (or $R^{nf}$).

### 4.5 Loss Function

#### 4.5.1 Trimap Loss

To learn the trimap of the foreground, we adopt the cross entropy loss on the semantic classification probabilities $s$:

$$\mathcal{L}_{sem}(s) = -\sum_{c=1}^{3} s_{GT}^c \log s^c, \tag{7}$$

where $s^c \in [0, 1]$ is the classification probability of $c$-th class in $s$ and $s_{GT}^c \in \{0, 1\}$ is the corresponding label. $s_{GT}^c$ is generated by the dilation and erosion operation on the ground-truth alpha.

#### 4.5.2 Alpha Loss

Before calculating the alpha loss, we convert the trimap prediction $t$ to the corresponding binary class masks of the foreground, background, and unknown region, denoted as $\{t^{fg}, t^{bg}, t^{un}\}$. Then, the two alpha losses, calculated on both unknown region ($\mathcal{L}_{detail}$) and the whole image ($\mathcal{L}_{alpha}$), are given by the following equations.

$$\mathcal{L}_{detail}(d, t^{un}) = t^{un}\{\|d - \alpha_{GT}\|_1 + \|\nabla d - \nabla \alpha_{GT}\|_1 + \sum_{k=1}^{5} \|Lap^k(d) - Lap^k(\alpha_{GT})\|_1\}, \tag{8}$$

$$\mathcal{L}_{alpha}(\alpha) = \|\alpha - \alpha_{GT}\|_1 + \|\nabla \alpha - \nabla \alpha_{GT}\|_1 + \sum_{k=1}^{5} \|Lap^k(\alpha) - Lap^k(\alpha_{GT})\|_1, \tag{9}$$

where $d$ is the predicted detail map, $\alpha_{GT}$ is the ground-truth alpha, and $\alpha$ is the alpha prediction given by $\alpha = d \cdot t^{un} + t^{fg}$. In addition to directly use the L1 loss, we also use the Sobel gradient operator ($\nabla$) and Laplacian pyramid ($Lap^k(\cdot)$) loss (Hou & Liu, 2019) to get better results.

#### 4.5.3 Foreground Loss and Re-construction Loss

To learn the foreground color prediction, we adopt L1 loss on the foreground prediction $F$ where $\alpha_{GT} > 0$:

$$\mathcal{L}_{fg}(F) = \|(\alpha_{GT} > 0) \odot (F - F_{GT})\|_1, \tag{10}$$

where $\odot$ is the Hadamard product.

We also use alpha prediction to re-construct the image using Eq. (1) and take the L1 loss when training with the synthetic data:

$$\mathcal{L}_{recons}(\alpha) = \|\alpha \cdot F_{GT} + (1 - \alpha) \cdot B - I\|_1, \tag{11}$$

where $B$ is the background for compositing to the training data, and $I$ is the original image.

#### 4.5.4 Overall Loss

We first train the Coarse Matting Network using following loss:

$$\mathcal{L}_{CMN} = \mathcal{L}_{sem}(s_c^{nf}) + \mathcal{L}_{detail}(d_c^{nf}, t_c^{nf(un)}) + \mathcal{L}_{alpha}(\alpha_c^{nf}) + \mathcal{L}_{fg}(F_c^{nf}) + \mathcal{L}_{recons}(\alpha_c^{nf}), \tag{12}$$

where $c$ is the down-sampling scale in CMN.

Once the training of CMN converges, we attach BMN to the CMN and train the entire model jointly using:

$$\mathcal{L}_{BMN} = \mathcal{L}_{CMN} + \mathcal{L}_{detail}(d^{nf}, t^{nf(un)}) + \mathcal{L}_{alpha}(\alpha^{nf}) + \mathcal{L}_{fg}(F^{nf}). \tag{13}$$

Table 2: Quantitative results on the synthetic data on our dataset (testing set). Best results under dynamic background are marked in blue, while those in green are the best under static backgrounds. A '†' indicates the model is retrained using our mixed training dataset. (BG=Background, FG=Foreground)

| BG Type | Input Image Type | | Method | Evaluation Metrics | | | |
|---|---|---|---|---|---|---|---|
| | | | | MSE↓ | SAD↓ | Grad↓ | Conn↓ |
| - | Single FG image | | MODNet† (Ke et al., 2022) | 1.751 | 4744.656 | 4188.432 | 3491.866 |
| | | | SHM† (Chen et al., 2018) | 3.289 | 5931.676 | 5462.052 | 5287.020 |
| | | | LF† (Zhang et al., 2019) | 3.504 | 6582.503 | 5911.190 | 5445.151 |
| | | | HATT† (Qiao et al., 2020) | 3.651 | 6892.154 | 6529.803 | 5792.352 |
| | | | RVM† (Lin et al., 2022) | 1.981 | 4270.903 | 7097.109 | 4133.624 |
| | | | SGHM† (Chen et al., 2022b) | 1.830 | 3859.964 | 4160.247 | 3837.194 |
| | | | PPM† (Chen et al., 2022a) | 2.891 | 5726.036 | 4793.306 | 4907.765 |
| | | | GFM† (Li et al., 2022) | 2.265 | 5415.296 | 4674.963 | 4321.387 |
| | | | MAM† (Li et al., 2024c) | 2.374 | 4480.037 | 3959.617 | 2202.677 |
| | | | SmartMat† (Ye et al., 2024) | 1.034 | 2450.608 | 4019.710 | 1753.940 |
| | | | P3M-Net† (Ma et al., 2023) | 1.733 | 3034.637 | 5130.156 | 2828.504 |
| Static | FG | BG | BGMv2 (Lin et al., 2021) | 0.396 | 1112.788 | 1851.081 | 992.213 |
| | Flash | No-flash | Ours | 0.634 | 1767.632 | 3697.195 | 1589.147 |
| Dynamic | FG | BG | BGMv2 (Lin et al., 2021) | 115.7 | 116675.8 | 32762.20 | 116327.7 |
| | FG | BG | BGMv2† (Lin et al., 2021) | 1.212 | 2744.400 | 5314.358 | 2615.139 |
| | FG | Other FG | | 2.547 | 4415.687 | 7394.905 | 4352.937 |
| | Flash | No-flash | | 1.989 | 3768.433 | 6802.648 | 3701.530 |
| | Flash | No-flash | Ours | 0.652 | 1804.741 | 3742.230 | 1626.339 |

# 5 Experiment evaluation

## 5.1 Data Preparation

### 5.1.1 Data Synthesis

**Foreground Acquirement.** In addition to training with our own constructed dataset, we also adopt foreground samples in VideoMatte240K (Lin et al., 2021) dataset for training. For a foreground video in this dataset, we select 10 frames with a large and uniform time interval and adjust their V channel in the YUV space to enhance the brightness. It forms flash/no-flash image pairs with the remaining frames of the video. For implementation, we use the `adjust_gamma` function in `skimage.exposure` to adjust the V channel, where the gamma value is randomly selected in $[0.2, 0.5]$. Although the pseudo flash image generated in this way is far from the real flash image, our model has no selectivity for the enhancement of foreground object color intensity, these images can effectively prevent the model from over-fitting with a small amount of data. Our model is trained on the mixed dataset of our real flash/no-flash image samples and pseudo flash/no-flash image samples from VideoMatte240K.

**Composite with Backgrounds.** Following Eq. (1), we can build flash/no-flash image pairs by compositing flash/no-flash foregrounds and their corresponding alpha matte onto dynamic background pairs under the same scene. During training, we first pair flash/no-flash foregrounds of the same person one by one. Then we select two background images of the same scene with a random interval from all the background images as a dynamic background pair. To enhance the robustness of our model to static backgrounds, we replace a certain dynamic background pair $(B_i, B_j)$ with a static background pair $(B_i, B_i)$ with a probability of 0.4, where $(B_i, B_j)$ represents two different background images but from the same scene and $(B_i, B_i)$ represents two identical background images. The background pair is composited with the previous foreground pair to obtain flash/no-flash image pairs for training.

**Test Set Building.** To build the test set, we composite pairs of flash/no-flash foreground with pairs of pre-randomly scrambled dynamic background images alternatively. A flash/no-flash foreground pair is composited with five different dynamic background pairs to form the test set.

**Adapt to Real-world Domain.** The model trained only on the synthetic data often suffers from the domain shift problem, *i.e.*, the model has a poor performance on real-world data. This is mainly due to

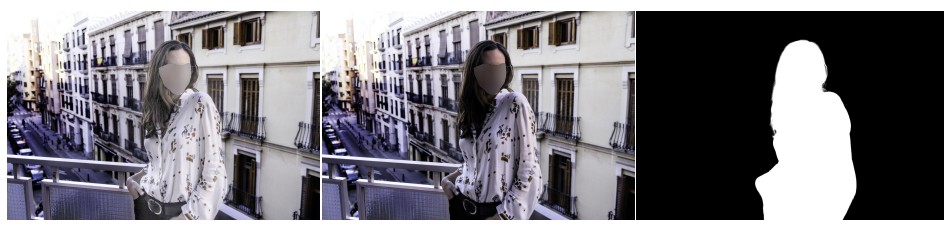

| Pseudo flash image | No-flash image | GT alpha |

Figure 8: Example of generated pseudo real-world flash/no-flash samples from PM-10K (Li et al., 2022).

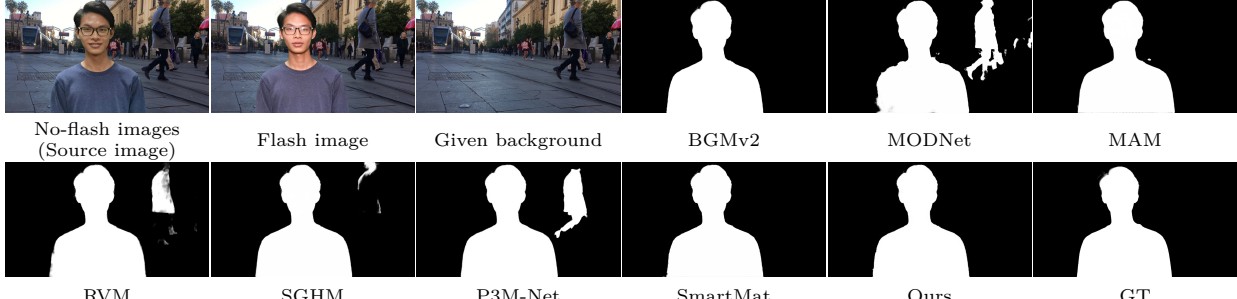

Figure 9: Qualitative results on the synthetic data with static backgrounds.

the compositional artifacts and invariable texture of these samples, leading to over-fitting of the model. To eliminate such a domain gap, in addition to introducing some data augmentation methods, we directly use PM-10K, a real-world portrait matting dataset proposed by Li et al. (2022) to transfer our model to the real-world domain. We use the same manner discussed above to generate the corresponding pseudo flash image for every real-world training sample in PM-10K dataset. Examples of generated training samples are shown in Fig. 8. However, we only brighten the foreground part of each sample by taking the weighted average of brightened image and the original image. The weight mask is obtained by applying erosion operation and averaging filter of the GT alpha matte.

### 5.1.2 Data Augmentation for Training

Moreover, we do the following augmentation operations for each pair of composited flash/no-flash image pairs. The operations include affine transformations, horizontal flipping, brightness, hue, and saturation adjustments on both foreground and background. We also apply blur, sharpening on the foreground randomly. The foreground is resized randomly and also placed to a random position to accommodate the perturbations caused by the movement of characters in the real world. After generating the training samples, each batch of images is cropped to a random height and width between 960 and 1600 to adapt the model to inputs of different resolutions.

### 5.2 Implementation Details

We use DeeplabV3 (Chen et al., 2017) weights pre-trained on Pascal VOC 2012 (Everingham et al., 2010) for semantic segmentation to initialize the backbone network and ASPP module, and adopt Adam as the optimizer. In the initial training of CMN, we set the down-sampling rate $c$ and batch size to 0.25 and 8 respectively. To learn the trimap prediction, the kernel size of erosion-dilation operation in generating GT trimap and $t_c^f$ is set to be $25 \times 25$. In FCM, we set $L_{nf} = L_f = 2$. And the learning rates are set to 5e-5 for the backbone, and 1e-4 for ASPP, FCM, and three decoders. In the joint training of CMN and BMN, the batch size is 4 and the learning rates of the backbone, ASPP, FCM, decoders, and BMN are set to 1e-5, 1e-5, 1e-5, 2e-5, 3e-5, 3e-4 respectively. In CMN training, we use a single Nvidia GeForce RTX 3090 to accelerate training. In the joint training, two Nvidia GeForce RTX 3090 cards are used for distributed training. For inference and following evaluation experiments, we set $c' = 20$, $r = 5$, $\delta = 0.7$, and $q = 5$ in the flash guided trimap prediction process. We also use Segment Anything Model (SAM) (Kirillov et al., 2023) as the pretrained segmentation model.

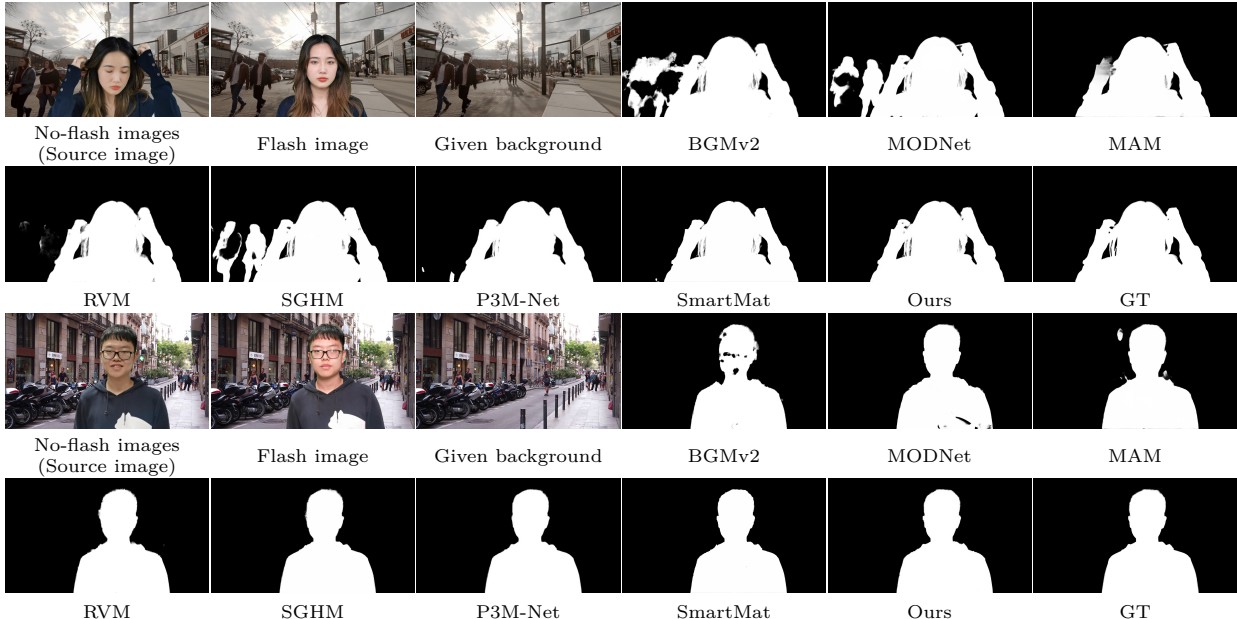

Figure 10: Qualitative results on the synthetic data with dynamic backgrounds.

## 5.3 Evaluation Metrics

We use four common metrics proposed in Rhemann et al. (2009) to measure the quality of the alpha matte prediction with ground-truth alpha matte, including mean square error (MSE), the sum of absolute difference (SAD), the gradient difference (Grad) and the connectivity difference (Conn). All MSE values are scaled by $10^{-3}$.

## 5.4 Comparisons on Synthetic Data

We composite testing foreground videos and testing dynamic background videos of our dataset introduced in Sec. 3 to build the test set, and perform both quantitative and qualitative comparisons on it.

### 5.4.1 Quantitative Comparison

For quantitative comparison, we benchmark one background-based matting method (BGMv2 (Lin et al., 2021)), and several trimap-free matting methods including MODNet (Ke et al., 2022), SHM (Chen et al., 2018), LF (Zhang et al., 2019), HATT (Qiao et al., 2020), RVM (Lin et al., 2022), SGHM (Chen et al., 2022b), PPM (Chen et al., 2022a), GFM (Li et al., 2022), and P3M-Net (Ma et al., 2023). We also include two interactive matting methods, including SmartMat (Ye et al., 2024) and MAM (Li et al., 2024c). Since SmartMat supports automatic matting originally, we apply it in such mode directly. For MAM, since it requires user interaction as input, we utilize one of the state-of-the-art detection models (GDINO (Liu et al., 2025)) to obtain the bounding box of the foreground in the no-flash image (with the text input for GDINO set as "foreground person") as prompt for MAM for further alpha prediction. For a fair comparison, all methods are retrained on the same mixed dataset. For MAM, we follow its original training settings, which only train the matting components and let the SAM (Kirillov et al., 2023) backbone frozen. In addition, we retrain the BGMv2 with three types of input and evaluate them respectively, including with/without subject image pairs, different image pairs of the same subject, and flash/no-flash image pairs. The comparison with the BGMv2 is performed with both static and dynamic backgrounds. For static background, the input image pair has the same background. For dynamic background, the backgrounds of the input image pair are under the same scene but are different caused by dynamic changes. All the test samples for testing are resized to a resolution of $1280 \times 720$.

Table 3: Ablation study on the proposed FNF test set with dynamic backgrounds.

| Methods | Evaluate metrics | | | |
|---|---|---|---|---|
| | MSE↓ | SAD↓ | Grad↓ | Conn↓ |
| Basic | 1.874 | 5153.745 | 9239.828 | 5783.462 |
| +SepDecoder | 1.729 | 4426.630 | 7239.828 | 4330.583 |
| +BMN | 1.212 | 2744.400 | 5314.357 | 2615.139 |
| +$T^{nf}$ | 1.197 | 2617.628 | 5035.997 | 2380.563 |
| +$T^f$ | 0.836 | 2005.576 | 4257.153 | 1885.972 |
| +CBAM | 0.652 | 1804.741 | 3742.230 | 1626.339 |

(a) **Each component.** Progressively attaching each component can significant improve the performance, demonstrating importance of each module.

| $\delta$ | Evaluate metrics | | | |
|---|---|---|---|---|
| | MSE↓ | SAD↓ | Grad↓ | Conn↓ |
| 0.4 | 2.384 | 5769.857 | 6834.817 | 4743.609 |
| 0.5 | 1.214 | 2996.780 | 4970.955 | 2498.168 |
| 0.6 | 0.794 | 2257.089 | 4275.529 | 1841.292 |
| 0.7 | 0.652 | 1804.741 | 3742.230 | 1626.339 |
| 0.8 | 0.701 | 1932.605 | 3861.104 | 1801.177 |
| 0.9 | 0.893 | 2160.262 | 4342.540 | 2389.265 |

(b) **Flash ratio map truncation threshold $\delta$.** Using lower or higher $\delta$ will result in inaccurate $t_c^f$, thus degrade the performance.

| Kernel Size | Evaluate metrics | | | |
|---|---|---|---|---|
| | MSE↓ | SAD↓ | Grad↓ | Conn↓ |
| 15×15 | 0.649 | 1799.543 | 3767.761 | 1617.709 |
| 25×25 | 0.652 | 1804.741 | 3742.230 | 1626.339 |
| 35×35 | 0.698 | 2071.123 | 3868.911 | 1729.099 |
| 45×45 | 0.791 | 2105.520 | 4132.484 | 1966.840 |
| 55×55 | 0.878 | 2437.644 | 4519.305 | 2352.018 |

(c) **Kernel size of erosion-dilation for generating GT trimaps.** Larger kernel size of erosion-dilation will make the performance worse.

| Medium Filter Kernel Size | Evaluate metrics | | | |
|---|---|---|---|---|
| | MSE↓ | SAD↓ | Grad↓ | Conn↓ |
| 3 | 0.652 | 1804.810 | 3741.978 | 1626.490 |
| 5 | 0.652 | 1804.741 | 3742.230 | 1626.339 |
| 7 | 0.652 | 1804.740 | 3742.230 | 1626.338 |
| 9 | 0.652 | 1804.741 | 3742.231 | 1626.339 |

(d) **Kernel size of medium filter for generating flash trimap.** The effect of adopting different medium filter kernel on the performance is negligible.

| $q$ | Evaluate metrics | | | |
|---|---|---|---|---|
| | MSE↓ | SAD↓ | Grad↓ | Conn↓ |
| 3 | 1.199 | 2641.574 | 5496.697 | 2682.519 |
| 4 | 0.686 | 1819.145 | 3797.614 | 1655.620 |
| 5 | 0.652 | 1804.741 | 3742.230 | 1626.339 |
| 6 | 0.652 | 1804.742 | 3742.230 | 1626.340 |

(e) **Number of confidence points $q$ for generating flash trimap.** Smaller $q$ results in worse performance.

| Segmentation model | Evaluate metrics | | | |
|---|---|---|---|---|
| | MSE↓ | SAD↓ | Grad↓ | Conn↓ |
| SAM | 0.652 | 1804.741 | 3742.230 | 1626.339 |
| InterFormer | 0.684 | 1815.367 | 3801.018 | 1652.824 |

(f) **Pre-trained segmentation model for generating flash trimap.** Our method is not selective for different segmentation models.

The quantitative comparison result is shown in Tab. 2. According to the results, we found our model shows superior performance on dynamic scenes while slightly falling behind BGMv2 on static scenes. This is mainly because modeling static and dynamic scenes involves contradictory objectives: static scenes prioritize the detection of foreground inconsistency solely, while dynamic scenes require modeling high-level foreground correspondences under motion and pose variations in both foreground and background. It is also worth knowing that dynamic scenes are a generalization of static ones, and training for broader generalization often reduces overfitting to simpler tasks, leading to slightly lower performance on static cases. This phenomenon aligns with the bias-variance dilemma (Zou & Hastie, 2005; Kohavi et al., 1996; Geman et al., 1992), showing that improving generalization may sacrifice some task-specific accuracy. Despite this, our model demonstrates strong generalization, excelling in dynamic backgrounds and effectively handling both static and dynamic scenarios. This versatility is a key strength, even if it comes at a small cost in static performance compared to static-specific methods like BGMv2.

Moreover, when training with flash/no-flash image pairs using the architecture of BGMv2, it performs much worse than our model. This also justifies the rationality of our model architecture for predicting alpha matte under dynamically changing backgrounds using flash/no-flash pairs. Also, since the input of trimap-free based methods is only one single image and does not contain any priors, they do not perform well in some diverse backgrounds. The corresponding quantitative results also show their performance is weaker than our model in the test set. It is worth noting that our proposed method still outperforms MAM (Li et al., 2024c) on the dynamic scenes although it adopts SAM (Kirillov et al., 2023) as backbone. However, in addition to generating $t_c^f$, our model does not directly use any external segmentation model to solve the alpha matte of the no-flash image.

### 5.4.2 Qualitative Comparison

We also select several latest state-of-the-art trimap-free methods (MODNet (Ke et al., 2022), RVM (Lin et al., 2022), SGHM (Chen et al., 2022b), P3M-Net (Ma et al., 2023), SmartMat (Ye et al., 2024), MAM (Li et al., 2024c)) and BGMv2 (Lin et al., 2021) for qualitative comparison. When dealing with static background, as shown in Fig. 9, we can see the performance gap between BGMv2 and our model is only reflected in the details of the edges in static backgrounds. However, BGMv2 performs poorly when the background is not aligned due to objects moving or camera shaking, which is shown in Fig. 10. Most of trimap-free methods also suffer from interference from background salient objects, but our model is robust to dynamic backgrounds and also have well performance.

### 5.5 Ablation Study

### 5.5.1 Component Analysis

To demonstrate the effect of the proposed Foreground Correlation Module and Boundary Matting Network, we first define the baseline as a network that only includes the encoder, ASPP module, and a single decoder without down-sampling. This baseline takes $I^{nf}$ as input only, and we denote it as **Basic**. We further replace the single decoder with three separate decoders, denoted as **SepDecoder**. Then we progressively attach BMN with down-sampling before the encoder, and each branch of FCM to build the corresponding combinations. These combinations are all retrained on the mixed dataset, and quantitative evaluation tests are carried out on the test set under dynamic background. The quantitative results are shown in Tab. 4a. We can see that decomposing the matting problem into semantic and detail predictions separately can effectively improve the matting performance. The BMN can recover fine details to predict alpha matte with higher-quality in the original resolution. However, attaching $T^{nf}$ with only self-attention blocks has a minor performance improvement, since a single no-flash image does not contain any foreground cues in various dynamic scenes. We further start to utilize the guidance information from $I^f$ by taking $(I^{nf}, I^f, t_c^f)$ as input, and introduce $T^f$ and add cross-attention blocks in $T^{nf}$ for modeling the foreground correspondence, resulting in a significant improvement. On this basis, the model with CBAM module adopted before $T^{nf}$ can achieve better performance, showing the effectiveness of feature re-attention operation.

### 5.5.2 Hyper-Parameter Analysis

We also conduct ablation studies on various hyper-parameter aspects, including the truncation threshold $\delta$ of the flash ratio map (presented in Tab. 4b), and the kernel size of erosion-dilation for generating the ground truth trimap (shown in Tab. 4c).

For the truncation threshold $\delta$, we observed that using a smaller threshold value degrades performance, as it results in a ratio map containing more inaccurate regions with low confidence scores. Conversely, a larger $\delta$ overly truncates regions with higher confidence, leading to inaccuracies in mask prediction for the flash image and a deterioration in model performance.

For the kernel size for erosion-dilation, we observed that employing a kernel size smaller than $25 \times 25$ can lead to slight improvements in model performance. Conversely, using a larger kernel size introduces more unknown regions in the ground truth trimap, potentially covering the entire portrait area. This forces the detailed branch to learn semantic information, ultimately resulting in a decrease in model performance.

We also explore the effect of adopting different kernel size of the medium filter and different number of confidence points $q$ when generating the flash trimap $t_c^f$. The corresponding quantative ablation results are presented in Tab. 4d and Tab. 4e, respectively. We found that using smaller or larger filter kernels has minimal impact on performance. However, larger kernels increase computational cost due to higher sorting complexity, leading us to adopt a median filter kernel size of 5 in this study. Regarding the number of points $q$, smaller values may result in obvious performance degradation, as insufficient confident points may fail to fully represent the foreground in the flash image, leading to incomplete or inaccurate flash cues. Conversely, larger $q$ values have almost no impact on performance.

### 5.5.3 Pre-trained Segmentation Model

The flexible design of our method allows the pretrained segmentation model for generating flash trimap can be replaced with any other more lightweight model to generate accurate flash cues. Such a substitution would not impact the overall performance of our approach, ensuring robustness and adaptability across different model choices. To further verify this, we conducted an ablation study by replacing current SAM (Kirillov et al., 2023) with InterFormer (Huang et al., 2023), a lightweight, real-time interactive image segmentation method that supports point-prompt inputs to generate segmentation masks, similar to SAM. The experimental results (showed in Tab. 4f) demonstrate that our model's performance remains largely unaffected by this substitution. Moreover, the use of InterFormer (Huang et al., 2023) accelerates the pre-processing stage, further enhancing the practicality of our approach.

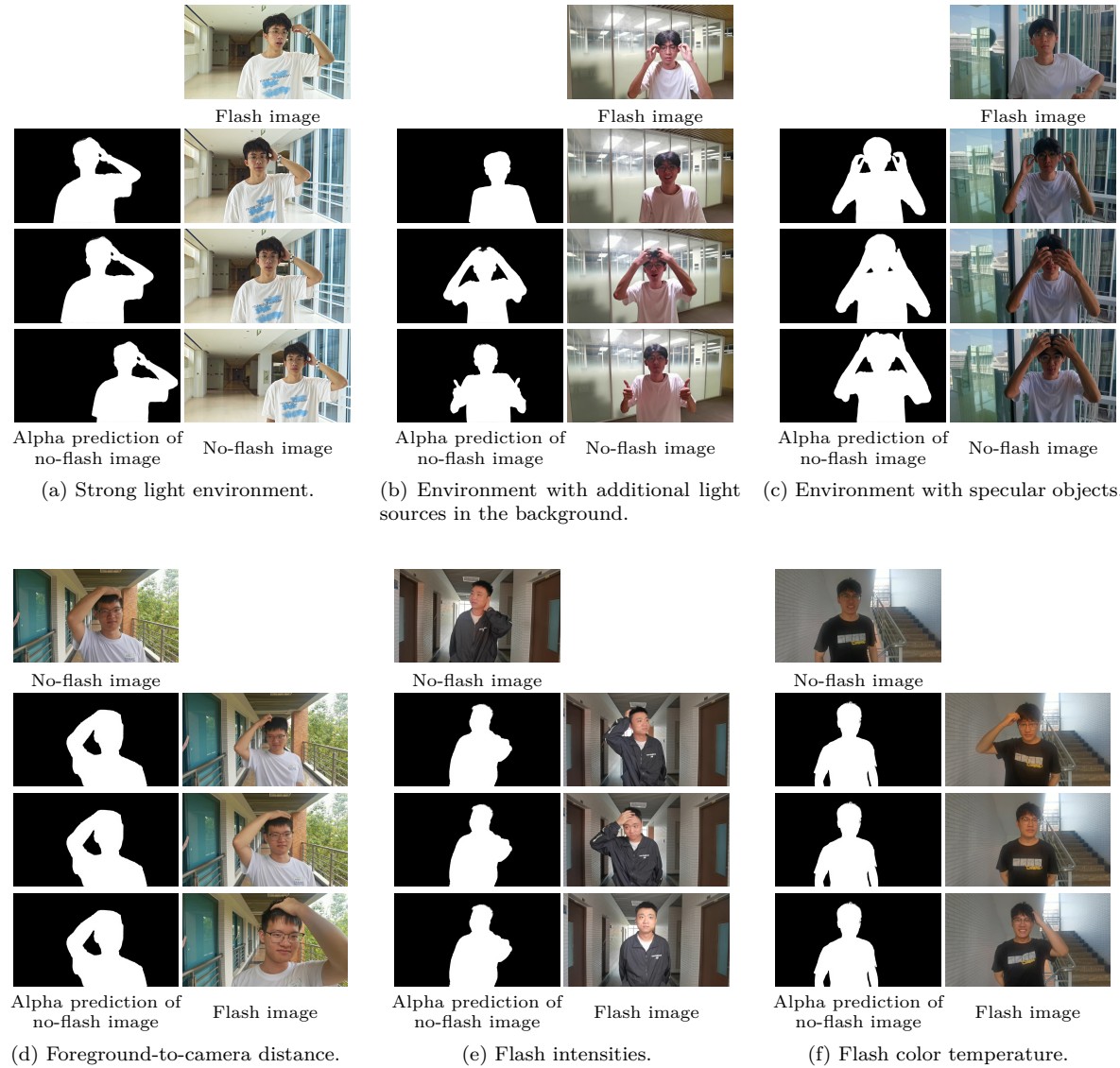

Figure 11: Evaluation in different light environment (a-c) and different flash conditions (d-f).

## 5.6 Environmental Robustness Evaluation

To assess the robustness of our model under varying light conditions, we conduct experiments in three specific lighting environments, which included:

1. A strong light environment (illustrated in Fig. 11a): Despite the significantly reduced difference between the flash image and the no-flash image in this scenario, our model could still extract correct flash cues and predict accurate alpha mattes for the no-flash image. This is attributed to the preprocessing of the flash ratio map, which includes truncation and median filtering operations.

2. An environment with additional light sources in the background (shown in Fig. 11b): Given that the flash ratio map is calculated based on the difference between the flash image and the no-flash image, it remains unaffected by background light sources under the flash photography assumption. Consequently, our model could successfully predict the correct alpha matte.

3. An environment with specular objects (depicted in Fig. 11c): When the light reflected by specular objects is not overly strong, the truncation and median filtering operations effectively filter out their impact, ensuring good model performance. However, in cases of excessively strong environmental light, such as a person facing the sun, where the flash is ineffective, our model may struggle to handle the situation. Additional discussions regarding this limitation can be found in Sec. 5.10.

In addition to assessing our model's performance under various lighting conditions, we also conducted evaluations under different flash conditions, encompassing:

1. Foreground-to-camera distance (illustrated in Fig. 11d): We ensured that the other two conditions remained unchanged while varying the foreground-to-camera distance. Despite the resulting reduction in flash differences between the flash image and the no-flash image, our model effectively utilized these differences as flash cues for prediction.

2. Flash intensities (shown in Fig. 11e): Similar to the previous scenario, we kept the other two conditions constant while adjusting flash intensities. Again, our model successfully harnessed the flash differences for accurate alpha matte prediction.

3. Flash color temperature (depicted in Fig. 11f): We illuminated the foreground with three different color temperatures (3,200K, 4,400K, and 5,600K from top to bottom) to acquire flash images. Despite the differences in color temperature, influenced by the flash, most foreground pixels exhibited an intensity increase. Since the flash ratio map is derived from pixel intensity enhancement, areas with higher confidence in the flash ratio map continued to represent the foreground. Consequently, our model accurately predicted the alpha matte of the no-flash image using flash images obtained under varying color temperatures. This demonstrates the robustness of our model to flash color temperature variations.

### 5.7 Evaluation on Real-world Data

We also capture some real-world videos using a smartphone containing flash and no-flash frames in dynamic scenes, and perform comparisons with BGMv2 and the four trimap-free methods.

Fig. 12 shows the comparison between our model and BGMv2. Since the basic assumption of BGMv2 is all pixels in the background are aligned, pixels changed in background will be considered as foreground, which causes the failure of BGMv2 in the dynamically changing regions. Moreover, the misjudgment of the semantic information also causes poor performance of boundary details prediction (*e.g.*, hands, eyeglasses) for BGMv2. This also highlights our method's core advantage, which lies in its robust handling of dynamic scenes, offering greater flexibility and usability compared to BGMv2, which is limited to static scenes with strict pixel alignment requirements.

Qualitative comparison with trimap-free methods on real-world data is also shown in Fig. 13. Although MODNet (Ke et al., 2022) can predict alpha matte with clearer details (*e.g.*, hair, eyeglasses), it usually failed in the semantic prediction. For RVM (Lin et al., 2022), it can predict the semantic information of the task well. However, since RVM up-samples the prediction results based on Deep Guided Filter (DGF) (Wu et al., 2018), resulting in unclear predictions for some regions such as glasses and fingers. SGHM (Chen et al., 2022b) and P3M-Net (Ma et al., 2023) also suffer from semantic prediction errors in some challenge cases, while our model is more robust and can produce alpha mattes with high-quality details.

### 5.8 Experiemnts in Low-light Environment

To further evaluate the effectiveness of our model, we conduct following experiments in the low-light environment. In such scenario, the contrast of foregrouds in the flash image and no-flash image will be very strong, which is beneficial for extracting robust guidance for the foreground from the flash image.

To verify this, considering there is no existing dynamic flash matting test set in low-light environment for evaluation, we collect a new benchmark by the following steps. We invited five participants to take the

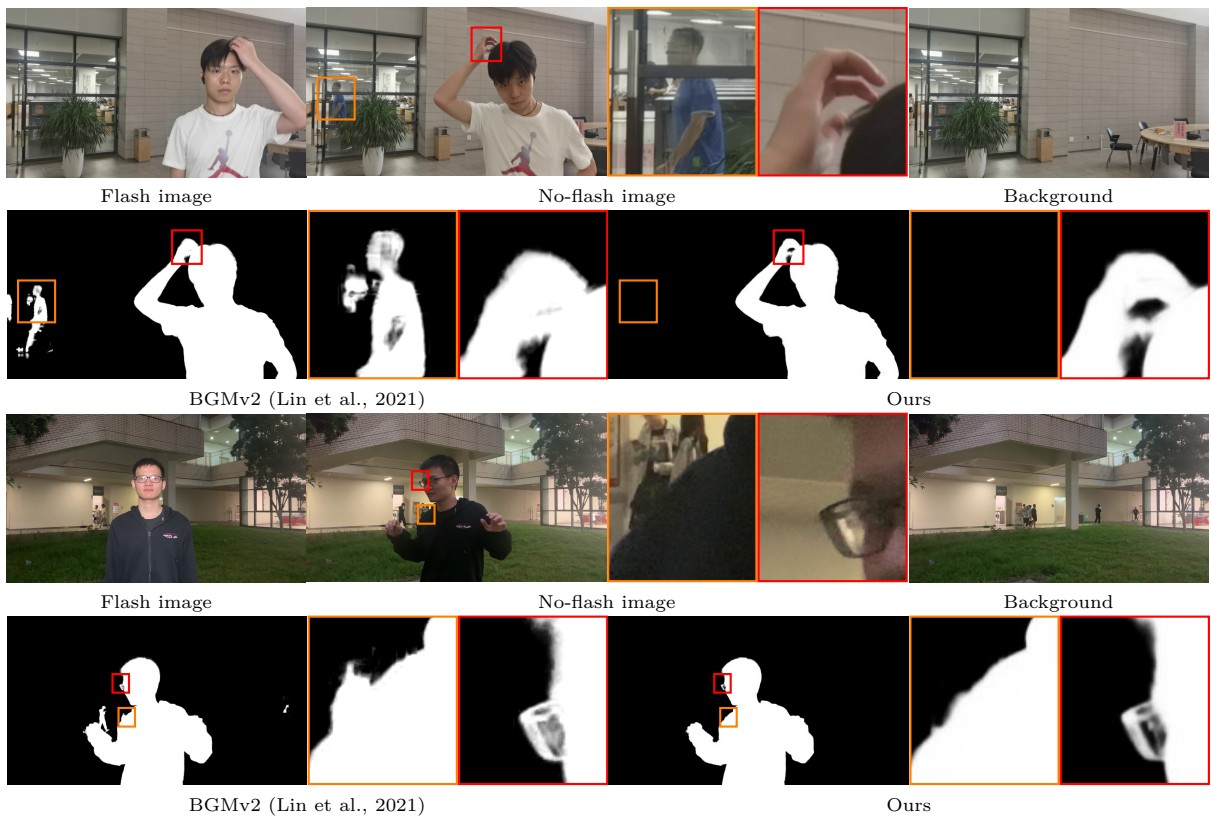

Figure 12: Comparison with BGMv2 (Lin et al., 2021) on real-world data captured by a smartphone. All the background provided to BGMv2 (Lin et al., 2021) is dynamic.

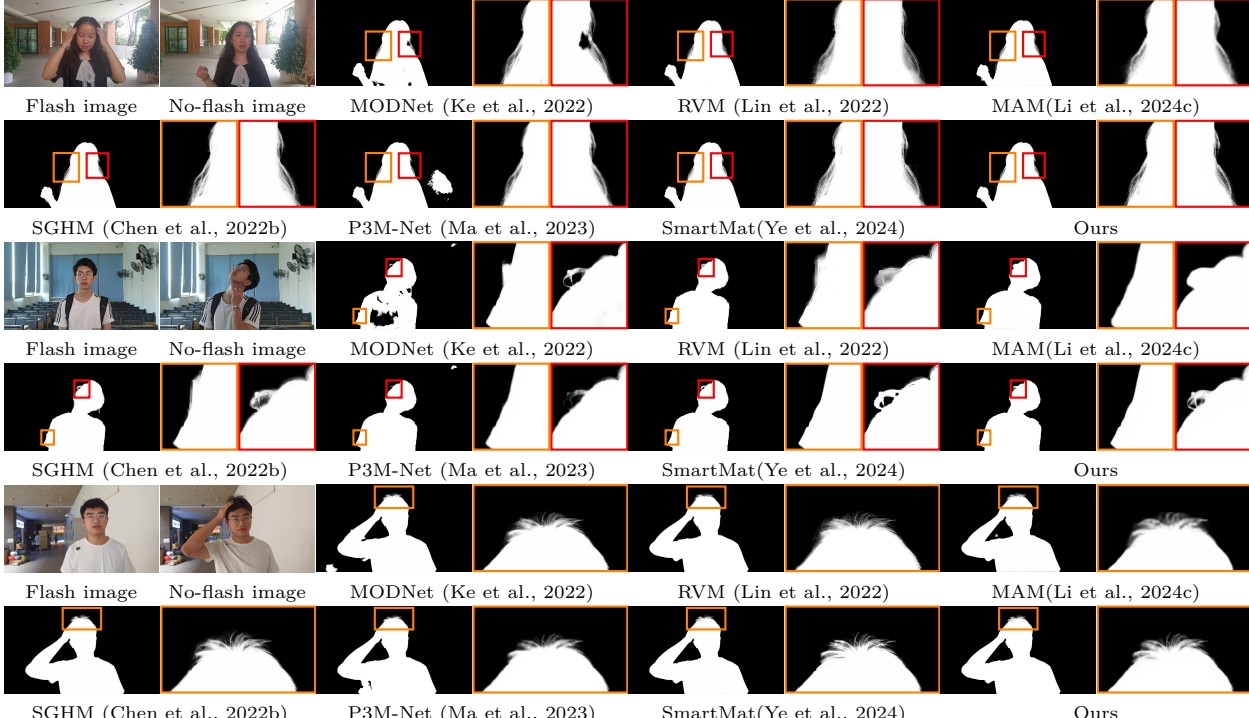

Figure 13: Comparison with trimap-free methods on real-world data captured by a smartphone.

Table 4: Quantitative comparison with baselines in the dark environment on our collected benchmark. Our method outperforms all the baselines in such low-light conditions.

| Methods | Evaluate metrics | | | |
|---|---|---|---|---|
| | MSE↓ | SAD↓ | Grad↓ | Conn↓ |
| BGMv2 (Lin et al., 2021) | 122.1 | 118376.4 | 19102.02 | 112306.5 |
| MODNet (Ke et al., 2022) | 21.58 | 23307.67 | 14334.61 | 17390.03 |
| RVM (Lin et al., 2022) | 1.658 | 4388.372 | 3992.540 | 3562.170 |
| SGHM (Chen et al., 2022b) | 1.014 | 2547.016 | 2652.587 | 1940.017 |
| P3M-Net (Ma et al., 2023) | 1.258 | 2943.804 | 3147.441 | 2241.799 |
| MAM (Li et al., 2024c) | 2.157 | 5329.854 | 3131.071 | 4503.777 |
| SmartMat (Ye et al., 2024) | 1.251 | 3048.106 | 5870.643 | 2322.956 |
| Ours | **0.863** | **1908.671** | **2174.382** | **1763.055** |

totally ten video clips in low-light environment using a mobile phone as camera. All the clips was captured in different outdoor scenes in evening. We also used a hand-held lamp to light the person intermittently during video recording process for further acquiring flash frame. We also ensured the camera parameters remain the same (*e.g.*, ISO, shutter speed) during taking the video in a single scene. For each video, we selected five representative frames from both the flash and no-flash sequences, and manually annotated GT alpha matte for them using Adobe Photoshop. Finally, we build a benchmark with 50 flash/no-flash image pairs for evaluation in low-light environment.

We then tested both the baselines and our method on this benchmark. The corresponding quantitative results are shown in Tab. 4. We found that even without including low-light data during training, our method still exhibited significant advantages over the baselines. This is because the stronger foreground contrast in low-light conditions allows the flash image to provide more effective guidance for matting the no-flash image. We also showcase the qualitative comparison in Fig. 14. We can see that BGMv2 (Lin et al., 2021) and MODNet (Ke et al., 2022) failed to generalize to the low-light environment. Although other trimap-free baselines show good performance in such scenarios, they tend to predict more blurred alpha in the boundary regions, especially for SGHM (Chen et al., 2022b) and P3M-Net (Ma et al., 2023). SmartMat (Ye et al., 2024) can produce more accurate alpha but with more artifacts and noise-like patterns in some cases (*e.g.*, the first sample in Fig. 14). In contrast, our method can generate more accurate alpha mattes even the no-flash image is slightly degradated due to the low-light condition, which also demonstrates the robustness of our model in real-world scenarios. In future work, we plan to collect more data in low-light environments to further advance research on the flash matting problem.

## 5.9 Model Complexity Comparison

We conduct a model complexity comparison, which encompasses the number of parameters (M), computational complexity (GMac), and inference time (ms), as detailed in Tab. 5. For the backbones, we use MobileNetV2(Sandler et al., 2018) for MODNet (Ke et al., 2022), ViTAE-S (Xu et al., 2021) for P3M-Net (Ma et al., 2023), DINOv2-ViTS14 (Oquab et al., 2024; Dosovitskiy et al., 2021) for SmartMat (Ye et al., 2024), SAM-B (Kirillov et al., 2023) for MAM (Li et al., 2024c), and ResNet-50 (He et al., 2016) for the remaining models. To assess inference time, we resized the input image to $1280 \times 720$, and the reported inference times were obtained using an NVIDIA RTX A5000. We also report the complexity of the preprocessing stage of our model, which includes generating the flash ratio map and a segmentation process (using SAM-B (Kirillov et al., 2023)) to obtain $t_c^f$.

We report the computational complexity and inference time for our model in two distinct cases. The first case, denoted as **1$^{\text{st}}$ frame**, involves the model taking $I_c^f, I_c^{nf}, t_c^f, I^{nf}$ as input and working end-to-end. In the second case, denoted as **other frames**, the computation result of the flash image until $T^f$ is given, and only new no-flash frames are forwarded, significantly reducing computational costs. Comparatively, BGMv2 (Lin et al., 2021) and RVM (Lin et al., 2022) exhibit lower computational complexity and inference time. This can be attributed to several factors, such as BGMv2 using a single decoder for direct alpha

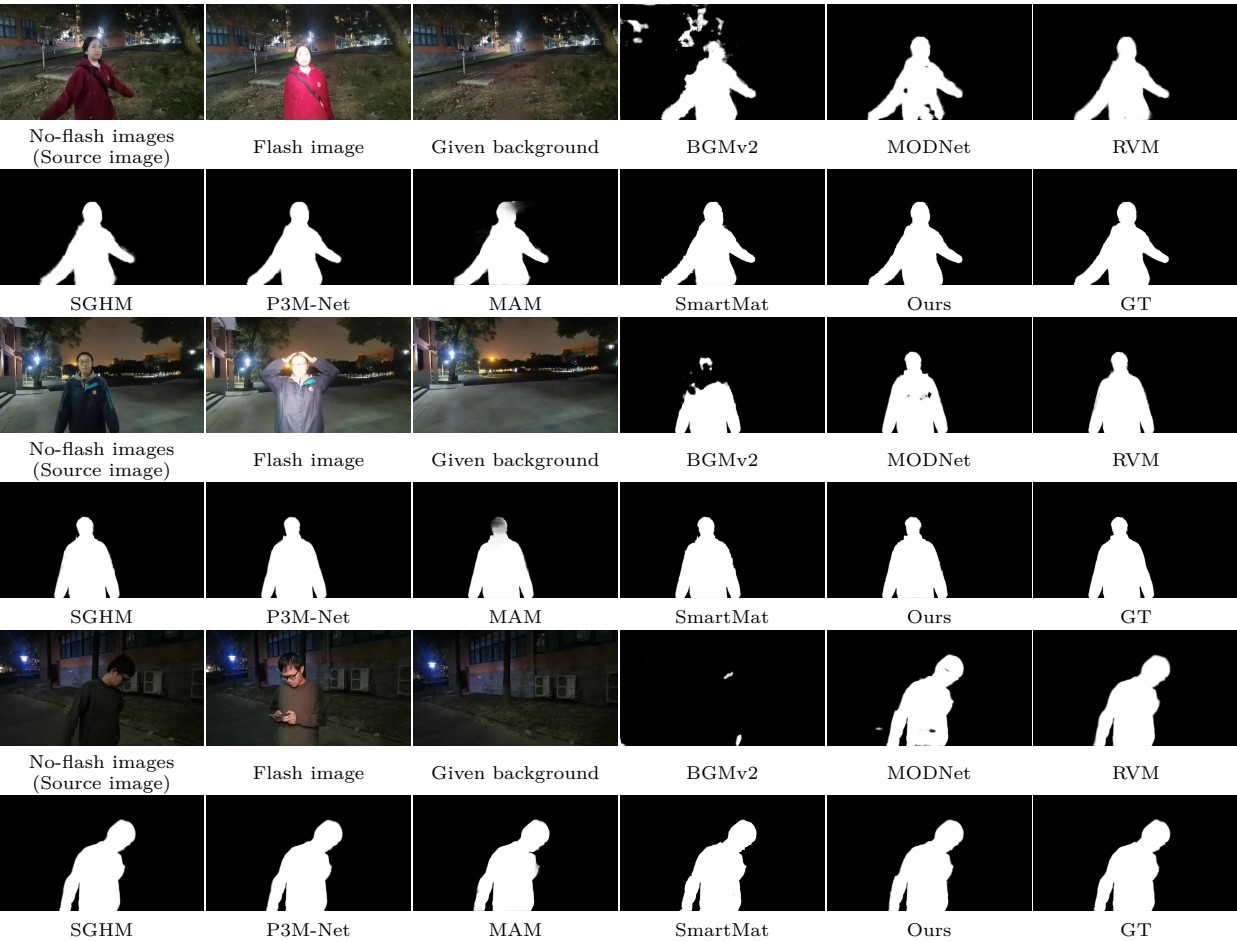

Figure 14: Qualitative comparison with baselines in the low-light environment. Zoom in for better viewing.

Table 5: Comparison of number of model parameters (M), computational complexity (GMac), and inference time (ms). For ours, (1$^{st}$ frame) and (other frames) represent forwarding the first no-flash frame and the flash image initially and forwarding other no-flash frames with given flash image features, respectively.

| Method | #Params (M) | Comp. (GMac) | Infer. (ms) |
|---|---|---|---|
| BGMv2 (Lin et al., 2021) | 40.25 | 17.47 | 13.27 |
| MODNet (Ke et al., 2022) | 6.49 | 64.95 | 27.90 |
| RVM (Lin et al., 2022) | 26.89 | 11.49 | 8.89 |
| SGHM (Chen et al., 2022b) | 40.25 | 25.81 | 25.96 |
| P3M-Net (Ma et al., 2023) | 39.48 | 300.39 | 36.79 |
| MAM (Li et al., 2024c) | 96.45 | 383.02 | 533.2 |
| SmartMat (Ye et al., 2024) | 26.89 | 228.73 | 38.96 |
| Preprocessing | 93.74 | 370.47 | 122.32 |
| Ours (1$^{st}$ frame) | 70.09 | 60.29 | 34.06 |
| Ours (other frames) | | 44.85 | 28.69 |

matte and foreground color prediction and RVM operating at lower resolutions and performing up-sampling through DGF (Wu et al., 2018) directly at the end of processing. Furthermore, SGHM (Chen et al., 2022b) only predicts alpha matte without foreground color, resulting in a lower computational complexity compared to our model. For MAM (Li et al., 2024c) and SmartMat (Ye et al., 2024), they have relatively higher GMac since they adopt ViT (Dosovitskiy et al., 2021) as backbone. Even if we choose the lightest version of SAM (SAM-B), MAM still suffers from high computational complexity and inference time.

Regarding our model, despite having a preprocessing stage with relatively higher complexity (since SAM is used) at the beginning, we observed a significant reduction in computational complexity and inference time when forwarding only new no-flash frames, while still maintaining competitive performance. This optimization is especially noticeable in the "only-no-flash forwarding" scenario, where our model's inference time closely aligns with that of MODNet (Ke et al., 2022) and SGHM (Chen et al., 2022b), despite our model's larger size. In practical application like video conference, since the preprocessing and "1$^{st}$ frame forwarding" stages typically only need to be performed once at the beginning, the overall efficiency of our model is comparable to other methods.

While it is true that our model boasts a larger number of parameters compared to other models, this is primarily attributed to our use of two separate ResNet-50 encoders that do not share weights, as well as the inclusion of the FCM module. Our ablation experiments, as detailed in Sec. 5.5, demonstrated the essential roles played by these two modules in establishing the connection between the flash image and the no-flash image foreground. These modules contributed significantly to the overall performance improvement of our model. In future work, we will explore the development of more lightweight modules for flash image encoding and foreground correction.

### 5.10 Extreme Cases Analysis

Our model performs reliably across a wide range of typical scenarios; however, failures may occur under specific, extreme conditions, which we systematically categorize into two aspects as following.

First, the flash photography assumption may violate in some rare scenarios where the flash/no-flash contrast is severely disrupted. These conditions lead to inaccurate flash ratio map estimations, resulting in unreliable matting outputs. For instance, when the foreground is outdoors under extremely strong sunlight (see Fig. 15a), the flash/no-flash photography assumption will be violated, since the flash difference between the flash image and the no-flash image becomes so minimal that no flash cues can be reliably detected. A potential solution to address this issue is the use of infrared light to illuminate the foreground, which will be explored in our future research. However, it is important to note that such light-related challenging scenarios are infrequent in typical environments, particularly indoors. These limitations do not compromise the performance of our model in applications such as video conferencing, live boardcasting, and virtual/augmented

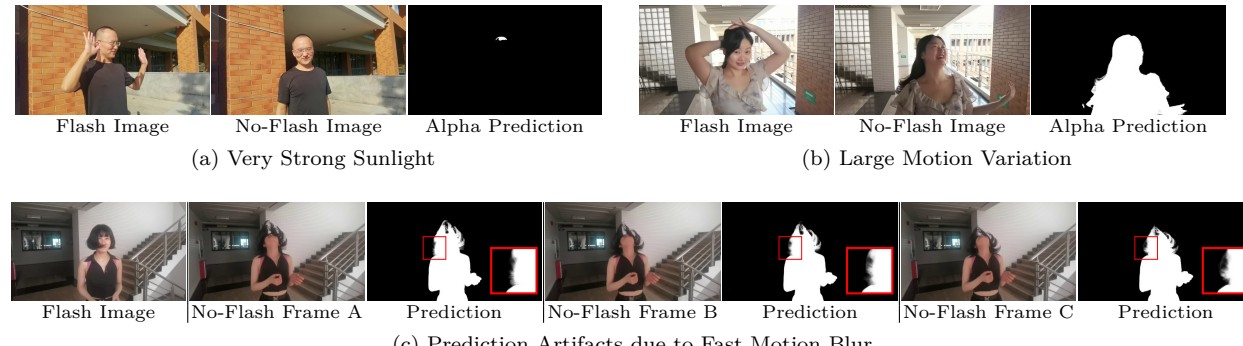

(a) Very Strong Sunlight

(b) Large Motion Variation

(c) Prediction Artifacts due to Fast Motion Blur

Figure 15: Extreme cases. Our method is susceptible to the effects of flash photography assumptions that are not met in such scenes with extreme light conditions, *e.g.*no enough flash contrast on the foreground in the environment with very strong sunlight (a). Other challenge cases such as large motion variation (b) or blurred image caused by extreme temporal conditions (*e.g.*, fast movement) can also cause our model to produce suboptimal results.

reality. Since modern imaging devices, including smartphones and webcams, are commonly equipped with built-in flash modules or external lighting aids, which are routinely used in professional and semi-professional setups to enhance visual quality. Our approach does not impose restrictive requirements such as continuous flash triggering or user calibration. Capturing a flash image is a one-time, automatic process that does not interfere with user experience. Compared to BGMv2 (Lin et al., 2021), which requires user intervention prior to each capture, our method offers improved usability and flexibility, making it more suitable for practical deployment.

Second, some extreme temporal conditions (e.g., fast movement), such as large motion variation or motion-blurred images, can also make our model to produce suboptimal results (see Fig. 15b and 15c). This limitation arises because our method processes frames independently, without explicit temporal modeling, making it sensitive to significant motion or image degradation in such rare scenarios. To address them for future research, we plan to integrate motion cues from flash/no-flash pairs and explore techniques such as a memory bank to explicitly model temporal information, further enhancing robustness for extreme pose variations while maintaining the model's strong performance in typical scenarios.

## 6   Conclusion

To solve the matting problem with dynamic backgrounds, this paper proposes a portrait matting model based on flash and no-flash images, which provides additional foreground priors to reveal the foreground objects. Our model is insensitive to dynamically changing backgrounds, and does not require that the movements of the foreground characters are exactly the same in the two images. Therefore, taking a flash image can map out multiple no-flash images, which is very practical. This paper also presents the first high-quality matting dataset of flash and no-flash single-person images, which provides an effective resource for subsequent research work on flash and no-flash matting work. Experiments show that our proposed method outperforms existing matting methods that do not require a tripartite map as a prior in dynamic backgrounds, showing the effectiveness of the model.

**Future Works.** Although our method shows strong generalization on both static and dynamic backgrounds, it still has slightly weaker performance in static scenes comparing with BGMv2 (Lin et al., 2021). Apart from the future work discussion in Sec. 5.10, we also provide two possible future directions to improve this. First, we can further integrate some pixel-flow estimation techniques, such as optical flow, into our foreground correlation modeling based on current flash cues, which can better capture subtle spatial and temporal relationships in static scenes. Optical flow can provide precise motion cues, enabling the model to distinguish static background elements from foreground objects more effectively. Another direction is to explicitly model static background information by introducing a background restoration subtask that

uses a generative inpainting prior, taking the flash cues as guidance. By training the model to reconstruct background regions, it can learn a robust representation of static scene features, thereby reducing artifacts and improving consistency in static matting tasks. Furthermore, considering the privacy-sensitive scenarios, we can also adapt the method to operate with infrared illumination, which is invisible to the human eye. We leave this promising direction for future research.

**Acknowledgments.** This research is supported by the Guangdong Natural Science Funds for Distinguished Young Scholars (Grant 2023B1515020097), Natural Science Foundation of Guangdong Province (2023A1515012894), Key R&D Project of Guangzhou Science and Technology Plan (2023B01J0002), National Natural Science Foundation of China (Grant No.: 62502117). And the National Research Foundation Singapore under its AI Singapore Programme (AISG Award No: AISG3-GV-2023-011), the National Research Foundation Singapore under the AI Singapore Programme (AISG Award No: AISG4-TC-2025-018-SGKR), the Ministry of Education, Singapore, under its Academic Research Fund Tier 2 (Award No. MOE-T2EP20125-0016), the Ministry of Education, Singapore, under its Academic Research Fund Tier 1 (Award No. MSS25C004), the National Natural Science Foundation of China (Grant No.: 62502117), the Shenzhen Natural Science Foundation (No. JCYJ20250604145532041), and the Lee Kong Chian Fellowships.

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
