# Let Your Light Shine: Foreground Portrait Matting via Deep Flash Priors
# – Supplementary Materials –

**Tianyi Xiang**        *tianxiang6-c@my.cityu.edu.hk*
*Department of Computer Science, City University of Hong Kong*
*School of Computing and Information Systems, Singapore Management University*

**Yangyang Xu**[†]        *xuyangyang@hit.edu.cn*
*School of Intelligence Science and Engineering, Harbin Institute of Technology (Shenzhen)*

**Qingxuan Hu**        *qxhbbxx@gmail.com*
*School of Computing, National University of Singapore*
*School of Computing and Information Systems, Singapore Management University*

**Chenyi Zi**        *barristanzi666@gmail.com*
*Department of Data Science and Analytics, HKUST (Guangzhou)*
*School of Computing and Information Systems, Singapore Management University*

**Nanxuan Zhao**        *nanxuanzhao@gmail.com*
*Adobe Research*

**Junle Wang**        *wangjunle@gmail.com*
*Tencent*

**Shengfeng He**[†]        *shengfenghe@smu.edu.sg*
*School of Computing and Information Systems, Singapore Management University*

**Reviewed on OpenReview:** *https://openreview.net/forum?id=vxUiVJp2eM*

## 1 Flash Quality Tolerability

Our proposed model imposes no hard restriction between flash/no-flash image pairs. That is, a single flash image can guide the matting of multiple no-flash images when the background scene only has natural dynamic changes, and the guidance information in regard to the matting of a single no-flash image provided by multiple different flash images is almost the same.

We conduct the following quantitative experiments to assess the robustness of our model, as presented in Tab. 1. Starting with the evaluation results on the current test set (referred to as **No replacement**), we proceeded as follows: 1) **Same identity (no-flash)**: We replaced the no-flash foreground with four different no-flash foregrounds, all from the same identity, for each flash/no-flash image pair and then evaluated the model. 2) **Same identity (flash)**: Similarly, we replaced the flash foreground with four different flash foregrounds, all from the same identity, for each pair and conducted the evaluation. 3) **Different identity (flash)**: We replaced the flash foreground with four different flash foregrounds, each from a different identity, for each pair and performed the evaluation. It is worth noting that all backgrounds remained unchanged.

Based on the results, we observed that replacing either the flash foreground or the no-flash foreground with the same identity has little effect on the model's performance. This suggests that the same flash image

---

[†]Corresponding authors.

can effectively guide the matting of multiple no-flash images with varying foreground poses and dynamic background changes. This practicality is especially relevant in video matting scenarios. Furthermore, our network appears to prioritize distribution differences over local pixel-wise differences. This characteristic enables our model to handle various inputs effectively. On the contrary, utilizing unpaired flash images significantly degrades model performance. This underscores the importance of using flash images and further highlights that our model exhibits relatively loose input restrictions and strong practicality.

Table 1: Quantitative results of our model with flash images replaced by the same or different identities.

| Image Replacement Methods | Evaluate metrics | | | |
|---|---|---|---|---|
| | MSE↓ | SAD↓ | Grad↓ | Conn↓ |
| No replacement | 0.652 | 1804.741 | 3742.230 | 1626.339 |
| Same identity (no-flash) | 0.667 | 1912.454 | 3849.037 | 1682.390 |
| Same identity (flash) | 0.652 | 1804.806 | 3741.998 | 1626.450 |
| Different identity (flash) | 59.37 | 78521.41 | 14330.61 | 49752.17 |

## 2    Robustness to Dynamic Backgrounds

Our model is insensitive to changes in dynamic backgrounds like pedestrian walking or camera movement. To verify this, for all flash and no-flash image pairs in the test dataset, we assign them background pairs with different degrees of changes.

We utilize RAFT (Teed & Deng, 2020) to estimate optical flow and measure the degrees of changes for background pairs. Given that optical flow predictions may be inaccurate for scenes with substantial changes, we implemented a frame interval strategy. When the frame number difference between two background frames was less than this interval, we calculated the optical flow directly between them. However, if the frame number difference exceeded the interval, we introduced an intermediate frame by selecting one at the end of the interval. Subsequently, we calculated two optical flows: one between the intermediate frame and the first frame and another between the intermediate frame and the second frame. These two optical flows were then vector-added to obtain the corrected optical flow between the initial two frames. In our experiments, we set the frame number interval to 30. Once we had the optical flow, which can be visualized as a vector field, we calculated the average length of all vectors in the field as the difference value for that background pair. We proceeded to sort all background pairs within the same scene in ascending order based on their difference values. Finally, we categorized them into five grades according to their ranking percentages: <20%, 20%~40%, 40%~60%, 60%~80%, and ≥80%, resulting in five distinct types of backgrounds with varying rates of change.

We conduct a quantitative evaluation of alpha prediction on the test set, considering five different grades of composited backgrounds, as presented in Tab. 2. Additionally, we provided some qualitative test results in Fig. 1. The results indicate that our model's performance remains consistent for backgrounds with varying degrees of variation. While there is a slight degradation in performance when the degree of variation is significant, these differences are generally minor. This observation underscores the adaptability of our model, as it does not exhibit strong selectivity based on the degree of natural dynamic changes in the background. Thus, our model can effectively handle matting in various dynamic scenes.

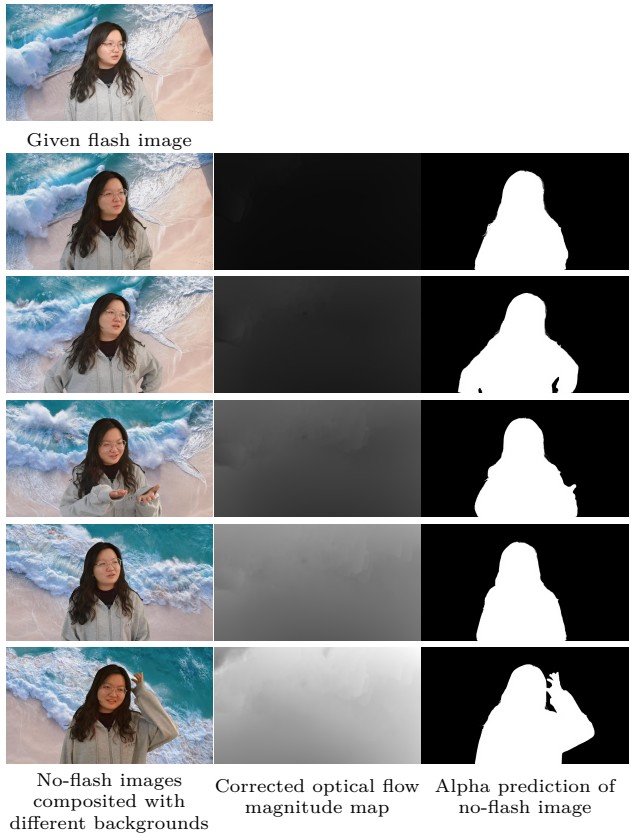

Given flash image

No-flash images composited with different backgrounds    Corrected optical flow magnitude map    Alpha prediction of no-flash image

Figure 1: Qualitative results of our model with different backgrounds in the same scene. The second column shows the magnitude map of the corrected optical flow of two backgrounds. The brighter the place in this map, the larger the magnitude of the optical flow vector here.

Table 2: Quantitative results of our model with different backgrounds in the same scene on our test set, where $\omega$ represents ranking percentage of difference of two backgrounds.

| Ranking Percentage $\omega\%$ | Evaluate metrics | | | |
|---|---|---|---|---|
| | MSE↓ | SAD↓ | Grad↓ | Conn↓ |
| $\omega < 20$ | 0.663 | 1851.338 | 3795.702 | 1655.836 |
| $20 \geq \omega > 40$ | 0.689 | 1879.646 | 3807.225 | 1689.918 |
| $40 \geq \omega > 60$ | 0.702 | 1894.698 | 3829.165 | 1701.664 |
| $60 \geq \omega > 80$ | 0.734 | 1904.221 | 3858.938 | 1722.240 |
| $\omega \geq 80$ | 0.758 | 1921.707 | 3887.520 | 1748.476 |

## 3 More Ablation Studies

### 3.1 Ablations about Settings

Here we conduct several ablations regarding the necessity of some settings of our model, including the feature fusion operation by FCM, flash cues, and applying box prompt for obtaining $t_c^f$. The results are shown in Tab. 3. We denote our default setting as **Default** in the table.

**Necessity of Flash/No-flash Feature Fusion.** To further investigate the effectiveness of our current design (*i.e.*, FCM-based feature fusion) in utilizing flash/no-flash cues, we conducted an ablation study in which we removed the Foreground Correlation Module (FCM) and used only the flash trimap and the no-flash image as inputs. According to the result (row 2 in Tab. 3), we found that the performance is degraded significantly since it is challenging to learn the matting for the no-flash image given the unpaired guidance

Table 3: Quantitative results of our model with different settings. Missing either flash feature fusion (row 2) or flash cues (row 3) leads to performance degradation. Adopting detection-based box prompt for acquiring $t_c^f$ (row 4) only provide negligible improvement.

| Settings | Evaluate metrics | | | |
|---|---|---|---|---|
| | MSE↓ | SAD↓ | Grad↓ | Conn↓ |
| Default | 0.652 | 1804.741 | 3742.230 | 1626.339 |
| w/o. $I^f$ and FCM | 1.273 | 2895.104 | 5476.496 | 2798.684 |
| w/o. $t_c^f$ | 0.954 | 2486.650 | 4627.361 | 2498.691 |
| w. box prompt | 0.651 | 1801.044 | 3738.101 | 1624.965 |

Table 4: Quantitative results regarding to ablation on training data and real flash effect. By adopting our proposed dataset only for training, the model can already achieve a relative remarkable performace. Introducing VM240K (Lin et al., 2021) and PM-10K (Li et al., 2022) further improve the performance and alleviate the real-composition domain gap issue. We also found training with pseudo flash images generated from no-flash images only show worse generalizability to the real-world dynamic flash matting problem.

| Training data | Evaluate metrics | | | |
|---|---|---|---|---|
| | MSE↓ | SAD↓ | Grad↓ | Conn↓ |
| FNF only | 0.854 | 2354.638 | 4195.965 | 2335.611 |
| + VM240K (Lin et al., 2021) | 0.684 | 1858.307 | 3765.889 | 1669.774 |
| + PM-10K (Li et al., 2022) | 0.652 | 1804.741 | 3742.230 | 1626.339 |
| FNF only (w/o. real flash) | 0.989 | 2529.935 | 4756.308 | 2538.973 |

with pose and location difference exists. This also indicates the effectiveness of the feature fusion operation in modeling the correspondence of the foregrounds to eliminate such motion-related perturbations.

**Necessity of Flash Cues.** The flash cues are used to extract $t_c^f$ via flash ratio map, which is essential to obtain more robust and complete foreground instance information in the flash image for guiding the matting of the no-flash image. To verify this, we only feed the CMN with $I_c^f$ and $I_c^{nf}$ without $t_c^f$, and let the network learn to determine the target foreground itself without any explicit guidance (row 3 in Tab. 3). We found there is a significant performance drop, since missing the flash cues will make the learning process confusing for the network to determine the target foreground instance in various dynamic scenes, especially if there is interference due to other persons with similar saliency in the background scene.

**Necessity of Box Prompt.** We also investigate the necessity of adopting an extra box prompt from the external detection-based model for obtaining $t_c^f$ in our model. In our default setting, we only use the points with higher confidence in the flash ratio map as a prompt to acquire $t_c^f$. Here, we attempt to introduce the representative detection model GDINO (Liu et al., 2025) to provide the extra box prompt for producing refined $t_c^f$. The input text of GDINO is set to "foreground person", and the output box and the original high-confidence points are combined as the prompts to the SAM (Kirillov et al., 2023) for further predicting the $t_c^f$ (row 4 in Tab. 3). However, we only found marginal improvement in the performance. This indicates the flash ratio map is already sufficient for extracting the foreground guidance of the flash image based on SAM.

## 3.2 Ablation on Training Data and Real Flash Effects

We train our model on three datasets, including our proposed FNF Matting dataset, VM240K (Lin et al., 2021), and PM-10K (Li et al., 2022). We conduct ablation studies to investigate the necessity of each of them for the training process. We start from our proposed dataset as the training data only and then add the other two datasets to the total training data progressively. The corresponding evaluation results are shown in Tab. 4. From the results, we found with our proposed dataset only, the model has shown good

enough performance (even better than most trimap-free methods), despite it still falling behind by the setting using VM240K. This indicates the effectiveness of our proposed dataset for training the model. In addition, although it is not very obvious that the contribution of the PM-10K dataset is reflected by the quantitative result, this dataset is nonnegligible since it helps our model generalize to the real-world domain effectively.

Moreover, we also conduct an ablation study to investigate the necessity of the real flash effects in the training data. Specifically, we trained our model using our FNF dataset but replaced the real flash images in each flash/no-flash pair with pseudo-flash images generated by adjusting the V channel of other no-flash images, following our previous setting. The test set remained unchanged, containing real flash images. The results (row 4 in Tab. 4) showed a significant performance degradation compared to training on our FNF dataset with real flash images (row 1 in Tab. 4), indicating that models trained on pseudo-flash images fail to generalize to real-world dynamic flash matting scenarios. This further underscores the necessity of using real flash effects for data collection.

## 4    More comparisons on real-world data

We provide additional qualitative results for comparison with BGMv2 (Lin et al., 2021) (Fig. 4) as well as other trimap-free methods (Fig. 2 and Fig. 3) using real-world data. These results demonstrate our model's ability to generate alpha mattes with intricate details across a wide range of real-world scenes.

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

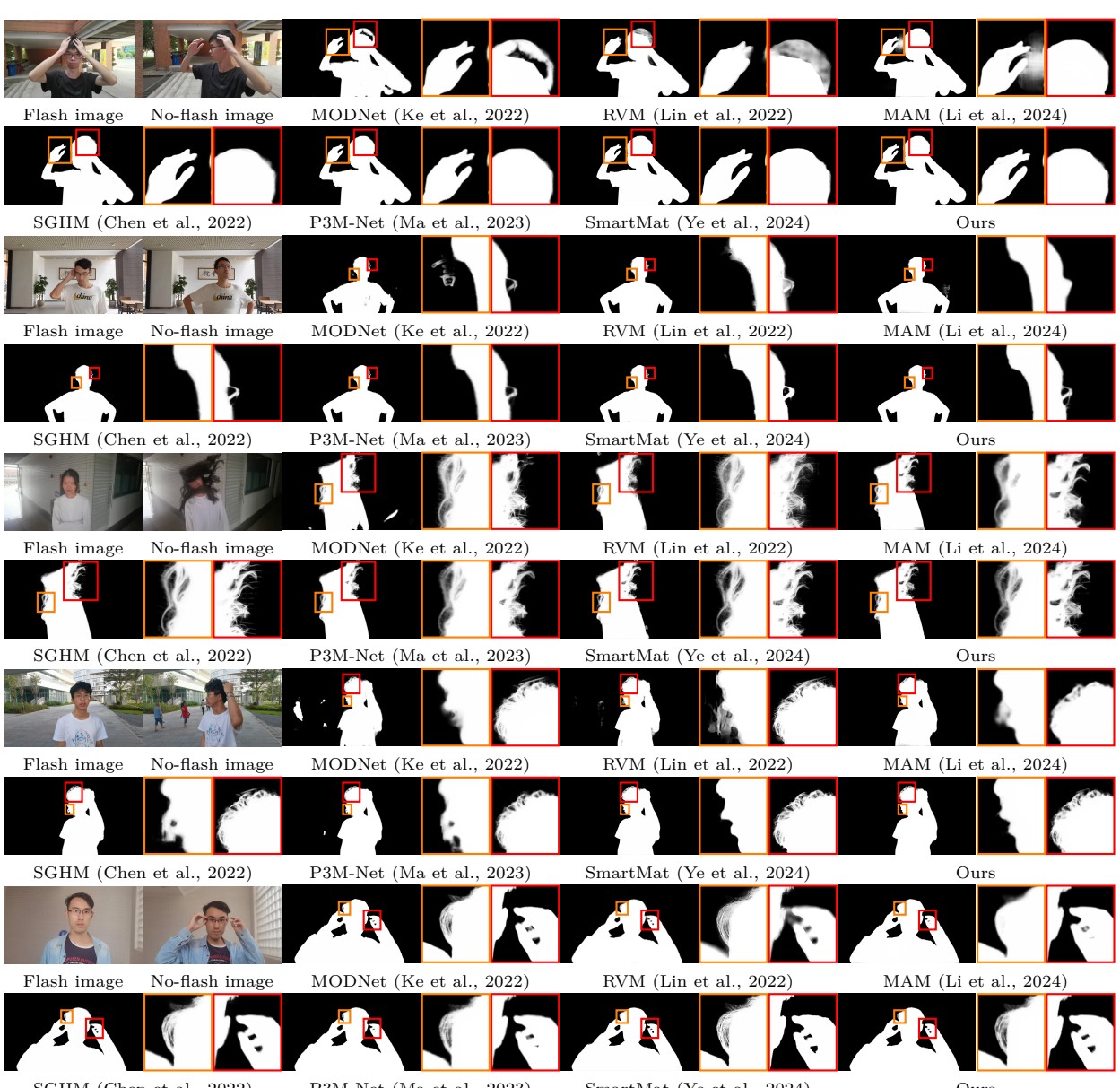

Figure 2: Comparison with trimap-free methods on real-world data captured by a smartphone.

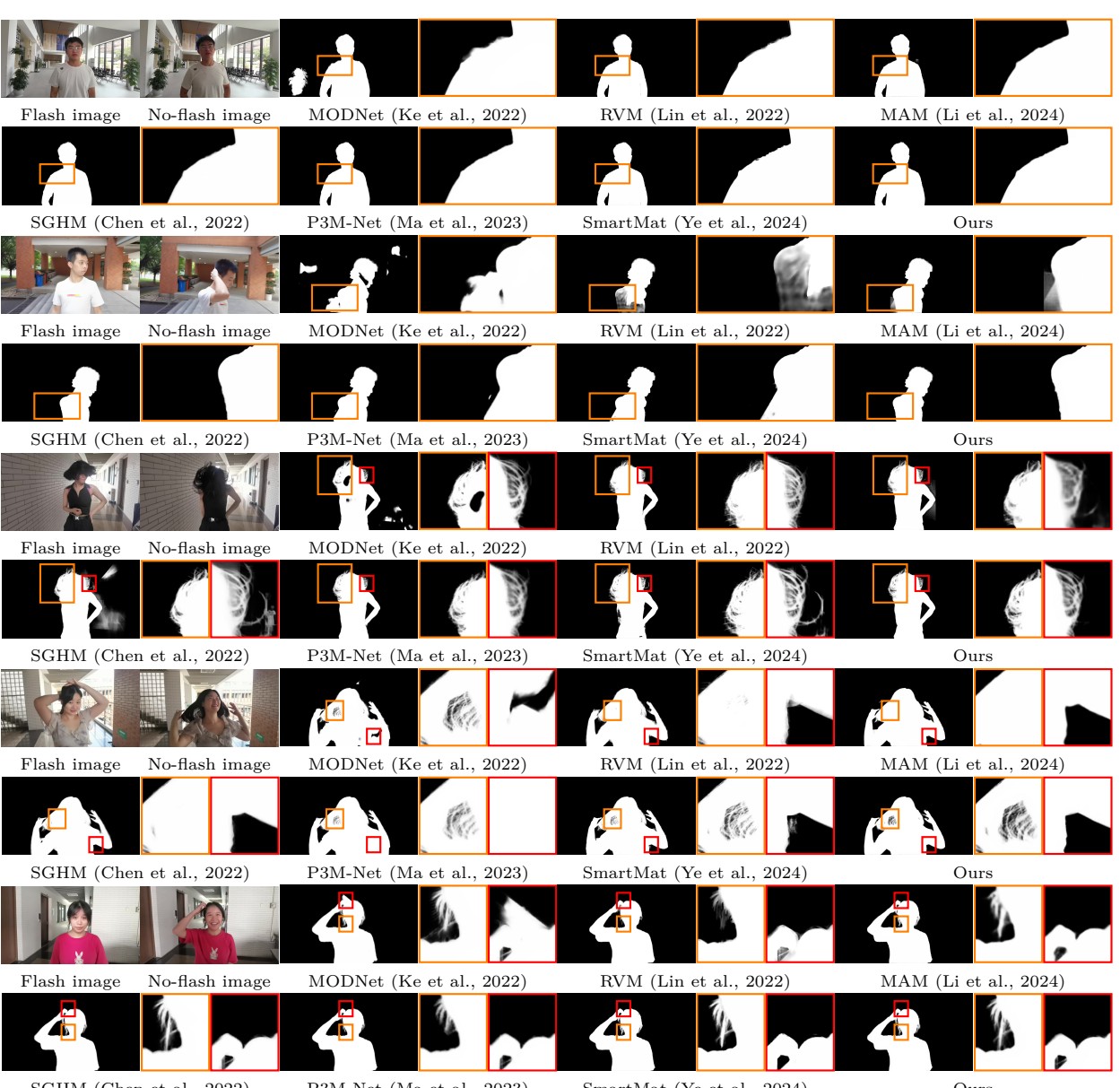

Figure 3: Comparison with trimap-free methods on real-world data captured by a smartphone.

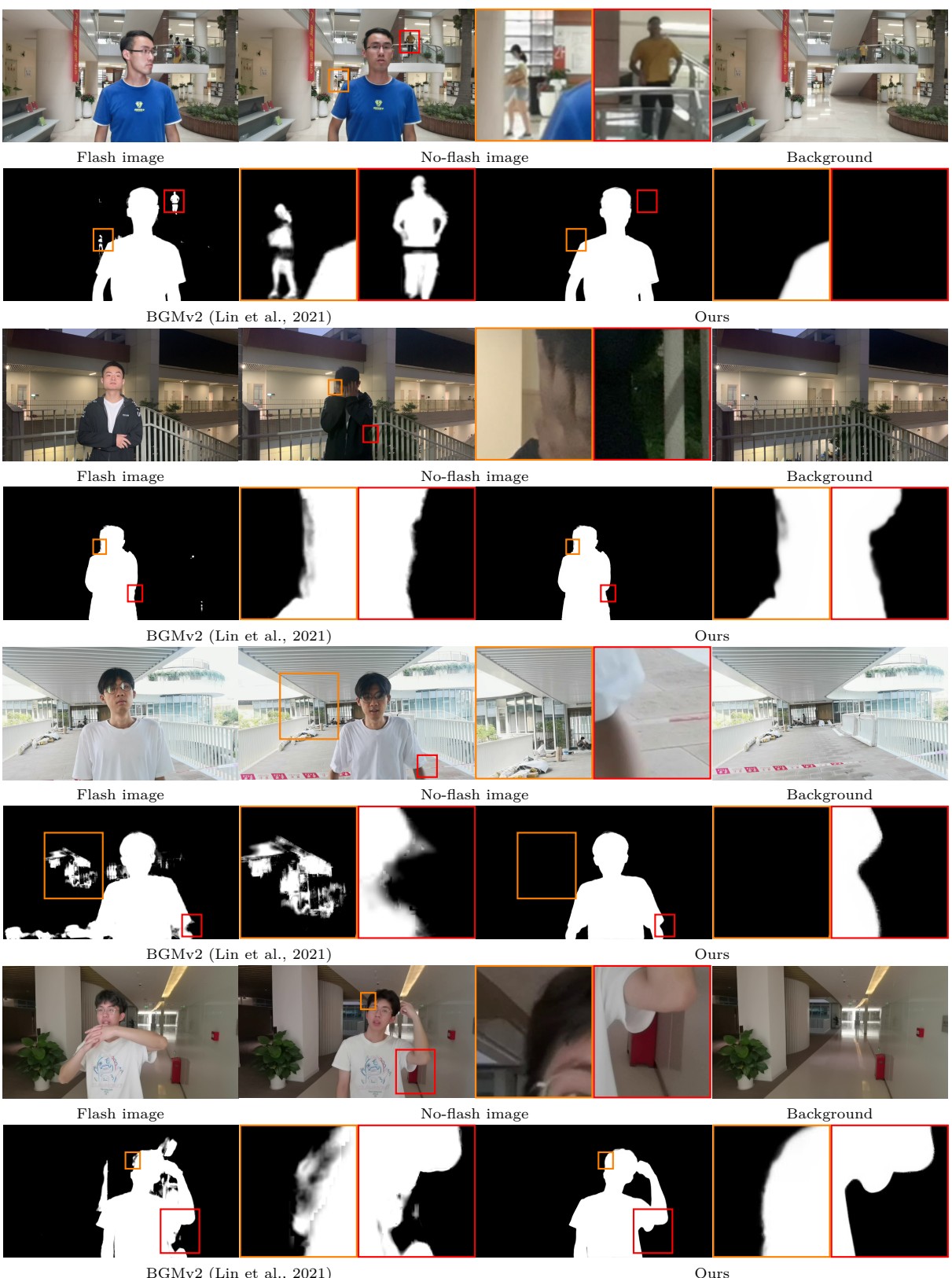

Figure 4: Comparison with BGMv2 (Lin et al., 2021) on real-world data captured by a smartphone. All the background provided to BGMv2 (Lin et al., 2021) is dynamic.