# OpenReview forum: "Let Your Light Shine: Foreground Portrait Matting via Deep Flash Priors"
_TMLR — Accepted by TMLR_

### Review · Reviewer_9Rpe · 2025-06-30

**Summary Of Contributions:**

To solve the problem of image matting with dynamic backgrounds, the authors explored a new approach to solve the image matting problem by leveraging flash prior knowledge to reveal the foreground. Specifically, the authors proposed a portrait matting model based on flash and non-flash images, which provides additional foreground prior knowledge to reveal foreground objects. The model is insensitive to dynamically changing backgrounds and does not require the foreground person to move exactly the same in both images. In addition, the authors constructed the first flash/no-flash portrait image matting dataset, which contains thousands of well-annotated alpha mattings. Finally, the authors verified the significant performance improvement of the proposed model in dynamic background scenes through experimental evaluation.

**Audience:**

Yes

**Broader Impact Concerns:**

The authors propose a novel solution for dynamic scene portrait matting, leveraging flash/no-flash image pairs. The proposed FNFNet demonstrates strong technical contributions, including an effective Transformer-based correlation module and a carefully designed two-stage architecture. The authors also construct the first large-scale flash/no-flash matting dataset, which will benefit future research. Overall, the method shows good practical application prospects and achieves state-of-the-art performance, especially in challenging dynamic backgrounds.

**Claims And Evidence:**

Yes

**Requested Changes:**

**(1) Data synthesis and consistency issues.**

Although the author manually annotates the foreground and alpha matte through Adobe Premiere, which is a method to ensure high-quality annotation, considering the large amount of data, this process is very cumbersome. It is recommended that the author add more descriptions on the annotation efficiency (such as whether semi-automatic auxiliary tools are used). In addition, for the process of synthesizing the foreground portrait with background videos of different resolutions and frame rates, the author should explain in detail its alignment strategy and visual consistency processing method to reduce the domain gap between the synthetic data and the real scene.

**(2) Insufficient discussion on the practical feasibility of Flash/No-flash dependence.**

The model is highly dependent on Flash/No-flash image pairs, but in the process of real video shooting, especially in privacy-sensitive scenarios such as meetings and live broadcasts, it is not realistic to frequently use flash. It is recommended that the author add analysis and discussion of deployment scenarios in the paper, such as whether there are alternatives or how to make adaptive adjustments when Flash images are missing, so as to improve the practicality of the method.

**(3) Paper formatting issues.**

The last reference in the second line of Section 2.2 seems to have a format error ("?"). Please check and correct the BibTeX or LaTeX reference format.

**(4) Analysis of the problem of slightly weaker performance in static scenes and discussion of optimization possibilities.**

Although the proposed method performs well in dynamic backgrounds, it is slightly inferior to BGM-V2 in static backgrounds. It is recommended that the author explore the reasons for this phenomenon and propose possible improvement directions, such as enhancing the segmentation accuracy of static scenes by fusing Flash and Background dual priors or designing a more fine-grained attention mechanism, thereby enhancing overall adaptability.

**Strengths And Weaknesses:**

**Strengths:**

(1) The authors propose a new "Flash/No-flash" image pair as a priori for foreground clipping, which cleverly avoids the limitations of the traditional "background image" method in dynamic backgrounds and camera movement scenes. It has practical application value and is especially suitable for actual scenarios such as video conferencing.

(2) The authors design a two-stage network (FNFNet), including a foreground correlation module and a boundary matting module, which effectively combines Flash priors with semantic information to predict high-quality alpha mattes. In particular, the Transformer-based structure is used to model the foreground correspondence under different lighting conditions, effectively enhancing the adaptability to dynamic scenes.

(3) The authors build a large-scale dataset containing 3025 high-quality annotated alpha mattes, covering a variety of backgrounds and character poses, filling the gap in public data, and with good statistical analysis and detailed description.

**Weaknesses:**

(1) The authors use Adobe Premiere to manually extract the foregrounds and alpha mattes, which seems to be a massive workload given the large volume of data. In addition, how the foreground portraits are blended with background videos of varying resolutions and frame rates to ensure that the resulting composite videos look visually coherent is also a concern. Improper settings may introduce a gap between the synthetic dataset and real-world data.

(2) The algorithm proposed by authors relies heavily on the quality of Flash/No-flash image pairs, but in reality most devices do not support frequent use of flash during video shooting, which may be inconvenient to use in privacy-sensitive scenarios (such as meetings and live broadcasts). Therefore, the reliance on Flash devices may limit the actual deployment of the algorithm.

(3) There appears to be a formatting error in the last citation on the second line of Section 2.2.

(4) The proposed method performs slightly worse than BGM-V2 in static scenes. Are there any potential improvements (e.g., improving the current model structure) that could address this issue in the future? The authors are encouraged to add relevant discussion on this.

---

> ### Author Response · Authors · 2025-08-07
> **Reply to Reviewer 9Rpe (1/2)**
>
> **W1**: *The authors use Adobe Premiere to manually extract the foregrounds and alpha mattes, which seems to be a massive workload given the large volume of data. In addition, how the foreground portraits are blended with background videos of varying resolutions and frame rates to ensure that the resulting composite videos look visually coherent is also a concern. Improper settings may introduce a gap between the synthetic dataset and real-world data.*
> **RC1**: *Data synthesis and consistency issues.*
>
> **A**: We would like to clarify that the data annotation process is not as labor-intensive as suggested. The collected video data features green-screen backgrounds, which significantly reduces the complexity of annotation compared to real-world matting scenarios. Furthermore, all video data was captured in a controlled setting using the same equipment, ensuring consistent lighting and camera parameters across the dataset. This uniformity allows us to configure Adobe Premiere with standardized parameters that are applicable to nearly all green-screen videos, enabling efficient batch processing to extract foregrounds and alpha mattes. In practical, setting annotation parameters for a single video takes less than 3 minutes, and annotate a single video only takes less than 10 minutes (including automatic annotation by Adobe Premiere and post-reviewing). The entire annotation process for the dataset was completed in 4 days with three skilled workers. We have already included the annotation details in Sec. 3.1 in the revised manuscript.
>
> To address the concern regarding the domain gap between synthetic and real-world data, we have implemented several measures to minimize artifacts and enhance visual coherence, as already detailed in Sec. 3.3 of the manuscript. Here we briefly summarize them as follow.
>
> - For resolution differences, we carefully curated high-resolution background videos to minimize resolution discrepancies, with the resolution distribution detailed in Sec. 3.2.2.
> - To address variations in noise distribution between foreground and background in synthetic data, we applied a denoising method before introducing controlled noise during training, effectively mitigating the impact of noise discrepancies.
> - We also incorporated artifacts through re-JPEG operations on composited data as part of our data augmentation strategy.
>
> Additionally, since both training and testing data consist of images, differences in frame rates between foreground and background videos are negligible. These strategies collectively ensure that the synthetic dataset closely aligns with real-world data, reducing the domain gap.
>
> **W2**: *The algorithm proposed by authors relies heavily on the quality of Flash/No-flash image pairs, but in reality most devices do not support frequent use of flash during video shooting, which may be inconvenient to use in privacy-sensitive scenarios (such as meetings and live broadcasts). Therefore, the reliance on Flash devices may limit the actual deployment of the algorithm.*
> **RC2**: *Insufficient discussion on the practical feasibility of Flash/No-flash dependence.*
>
> **A**: Thank for your comment. To clarify, although the flash image is required in our setting, our method does not require frequent use of flash. As demonstrated in Sec. 1 of the supplementary PDF, a single flash image can guide the matting process for multiple no-flash images within the same scene. The corresponding experiment shows that replacing a no-flash image with another from the same scene (maintaining the same subject identity but with varying poses or backgrounds) has no significant impact on model performance, indicating that our approach is not overly dependent on specific flash/no-flash pairings.
>
> In practical, the primary applications of matting, including video conferencing, film production, XR environments, and live streaming, generally permit the use of flash. Modern imaging devices, including smartphones and webcams, are commonly equipped with built-in flash modules or external lighting aids, which are routinely used in professional and semi-professional setups to enhance visual quality. Our approach does not impose restrictive requirements such as continuous flash triggering or user calibration. Capturing a flash image is a one-time, automatic process that does not interfere with user experience. Compared to BGMv2, which requires user intervention prior to each capture, our method offers improved usability and flexibility, making it more suitable for practical deployment.
>
> Finally, the proposed method can be adapted to operate with infrared illumination, which is invisible to the human eye. This suggests that our framework holds potential for privacy-sensitive scenarios and offers a promising direction for future research. We have included this discussion in Sec. 5.10 and Sec. 6 of the revised manuscript.

---

> ### Author Response · Authors · 2025-08-07
> **Reply to Reviewer 9Rpe (2/2)**
>
> **W3** & **RC3**: *There appears to be a formatting error in the last citation on the second line of Section 2.2.*
>
> **A**: We thank the reviewer for pointing out the formatting error in the last citation on the second line of Sec. 2.2. This issue has been corrected in the revised manuscript.
>
> **W4**: *The proposed method performs slightly worse than BGM-V2 in static scenes. Are there any potential improvements (e.g., improving the current model structure) that could address this issue in the future? The authors are encouraged to add relevant discussion on this.*
> **RC4**: *Analysis of the problem of slightly weaker performance in static scenes and discussion of optimization possibilities.*
>
> **A**: Thank you for the feedback. It is important to note that modeling dynamic and static scenes inherently involves conflicting objectives. In static scenes, the primary focus is detecting inconsistencies in the foreground, whereas dynamic scenes require accurately capturing high-level correspondences between foreground and background under motion and pose variations. These distinct requirements naturally lead to an asymmetric performance trade-off.
>
> Furthermore, dynamic scenes can be viewed as a generalization of static scenes. In machine learning, models trained under simpler assumptions (e.g., static backgrounds) often excel in specific tasks but may struggle in more complex scenarios due to overfitting to a limited problem scope. Conversely, models trained on dynamic scenes develop stronger generalization capabilities, which may result in slightly reduced performance on simpler tasks. This trade-off aligns with the well-established **bias-variance dilemma** [1,2,3]. A similar effect is observed in matting tasks; for instance, the SmartMatting model [4] demonstrates strong generalization across diverse matting tasks but slightly underperforms task-specific baselines due to its broader applicability.
>
> Our method exhibits a similar generalization effect, achieving superior performance in dynamic scenes—where adaptability is critical—while slightly underperforming BGMv2 in static scenes. This is a natural outcome of prioritizing generalization, which sometimes necessitates marginal trade-offs in specific scenarios. To address this limitation and enhance performance in static scenes while preserving generalization, we propose two promising directions for future work:
>
> - Incorporating Pixel-Flow Estimation into Foreground Correlation Modeling: Based on the flash cues, we can further integrate some pixel-flow estimation techniques, such as optical flow, into our foreground correlation modeling, which can better capture subtle spatial and temporal relationships in static scenes. Optical flow can provide precise motion cues, enabling the model to distinguish static background elements from foreground objects more effectively, thus improving matting accuracy in static settings without compromising dynamic scene performance.
>
> - Leveraging Generative Inpainting Priors with Background Restoration: Taking our flash cues as guidance, we can also explicitly model static background information by introducing a background restoration subtask that uses a generative inpainting prior. By training the model to reconstruct background regions, it can learn a robust representation of static scene features, thereby reducing artifacts and improving consistency in static matting tasks. This approach will complement our existing foreground-centric modeling and improve performance in static scenes.
>
> These strategies aim to address the lack of explicit static background modeling in our current approach, bridging the performance gap with BGMv2 in static scenes while maintaining our method’s strong generalization to dynamic scenes. The analysis of the phenomenon has been provided in Sec. 5.4.1. And we have also discussed the potential future improvement directions in Sec. 6 of the revised manuscript.
>
>
> ### Reference
> [1]	Geman, S., Bienenstock, E., Doursat, R.: Neural networks and the bias/variance dilemma. Neural computation 4(1), 1–58 (1992)
> [2]	Kohavi, R., Wolpert, D.H., et al.: Bias plus variance decomposition for zero-one loss functions. In: ICML, vol. 96, pp. 275–283 (1996)
> [3] Zou, H., Hastie, T.: Regularization and variable selection via the elastic net. Journal of the Royal Statistical Society Series B: Statistical Methodology 67(2), 301–320 (2005)
> [4] Ye, Z., Liu, W., Guo, H., Liang, Y., Hong, C., Lu, H., Cao, Z.: Unifying automatic and interactive matting with pretrained vits. In: CVPR, pp. 25585–25594 (2024)

---

### Review · Reviewer_WP5u · 2025-07-04

**Summary Of Contributions:**

In this paper, authors aim to propose a novel image matting method by revealing the foreground with flash priors, which could do not rely on the background image as reference. To this end, authors use a flash/no-flash image pair as a prior to highlight the foreground object. The proposed framework, FNFNet, is a two-stage cascaded network that uses a Transformer-based Foreground Correlation Module (FCM) to explore the relationship between the two differently illuminated images. To facilitate this research, authors also introduce the first large-scale flash/no-flash portrait matting dataset. In the experimental part, authors claim that their proposed method significantly outperforms other trimap-free matting methods in scenarios with dynamic backgrounds.

**Audience:**

Yes

**Broader Impact Concerns:**

There are not any further broader impact concerns.

**Claims And Evidence:**

Yes

**Requested Changes:**

There are several concerns expected to further demonstrate here:
1. The trimap prediction relies on a pre-trained SAM to generate the segmentation mask. How dependent is the final performance of proposed method on the quality of this external model? It would be better to provide an ablation study using a different and simpler segmentation model to investigate this dependency.
2. It would be better to include a comparison of model parameters and computational complexity (FLOPs, or inference time) in Table 2. This information is important for assessing the efficiency of FNFNet relative to the other methods, especially for potential real-time applications.
3. The pre-processing pipeline for generating the flash-trimap involves several hyperparameters (e.g., filter r, number of points q). Could the authors investigate the sensitivity of the final performance to these hyperparameters?

**Strengths And Weaknesses:**

There are several strengths here:
1. The core idea of using a flash/no-flash image pair as a prior is a interesting for image matting in dynamic scenes, addressing a key limitation of background-matting methods.
2. The introduction of a novel, large-scale dataset is highly valuable for promoting future research in community.
3. The experimental results seem to be good and convincing.


There are several weakness here:
1. The overall pipeline is quite complex, involving a multi-step pre-processing stage for trimap prediction before the main two-stage matting network.
2. The method's performance is potentially dependent on an external large pre-trained model (SAM), which might lead to extra params and inference time. I am not sure whether the proposed method could use other pretrained model to have similar performance.
3. The quantitative comparison in Table 2 lacks a discussion of model complexity, such as parameter counts or FLOPs, making it difficult to evaluate the method's efficiency.

---

> ### Author Response · Authors · 2025-08-07
> **Reply to Reviewer WP5u (1/2)**
>
> **W1**: *The overall pipeline is quite complex, involving a multi-step pre-processing stage for trimap prediction before the main two-stage matting network.*
>
> **A**: Thank you for your comment. The primary goal of our method is to extract complete, accurate, and robust flash cues for effective no-flash image matting. While the pre-processing stage involves multiple steps, these are essential and not overly complex. Specifically, the pre-processing operations are low-level (e.g., histogram computation, median filtering) and performed at a significantly reduced resolution (downsampled to $\frac{1}{20}$ of the original image). These operations do not constitute a bottleneck for inference speed. In practical, these low-level operations last less than 20 ms in total, using a common PC with a single A5000. Furthermore, in scenarios with stringent real-time requirements, such as video conferencing or live broadcasting, the flash cues derived from a single flash image during pre-processing can guide matting for multiple no-flash images in the same scene. This eliminates the need for real-time pre-processing, ensuring that our method maintains high efficiency without compromising performance.
>
> **W2**: *The method's performance is potentially dependent on an external large pre-trained model (SAM), which might lead to extra params and inference time. I am not sure whether the proposed method could use other pretrained model to have similar performance.*
> **RC1**: *The trimap prediction relies on a pre-trained SAM to generate the segmentation mask. How dependent is the final performance of proposed method on the quality of this external model? It would be better to provide an ablation study using a different and simpler segmentation model to investigate this dependency.*
>
> **A**: Thanks for pointing this out. The use of SAM [1] is solely to obtain precise and robust flash cues to guide matting for no-flash images. Even adopting the SAM [1], a single pre-process forwarding only lasts 0.1s using an A5000 GPU approximately. Besides, the flexible design of our method allows the SAM [1] can be replaced with any other more lightweight model to generate accurate flash cues. Such a substitution would not impact the overall performance of our approach, ensuring robustness and adaptability across different model choices. To further verify this, we conducted an ablation study by replacing SAM [1] with InterFormer [2], a lightweight, real-time interactive image segmentation method that supports point-prompt inputs to generate segmentation masks, similar to SAM.
>
> |  Pre-trained segmentation model | MSE $\downarrow$ | SAD $\downarrow$ | Grad $\downarrow$ | Conn $\downarrow$ |
> |:-----------:|:---:|:---:|:----:|:------:|
> |      SAM [1]     |  0.652 | 1804.741 | 3742.230 | 1626.339 |
> | InterFormer [2]  |  0.684  | 1815.367 |  3801.018 | 1652.824 |
>
> The experimental results demonstrate that our model’s performance remains largely unaffected by this substitution. Moreover, the use of InterFormer accelerates the pre-processing stage, further enhancing the practicality of our approach. These findings have been incorporated into the revised manuscript in Sec. 5.5.3.
>
> **W3**: *The quantitative comparison in Table 2 lacks a discussion of model complexity, such as parameter counts or FLOPs, making it difficult to evaluate the method's efficiency.*
> **RC2**: *It would be better to include a comparison of model parameters and computational complexity (FLOPs, or inference time) in Table 2. This information is important for assessing the efficiency of FNFNet relative to the other methods, especially for potential real-time applications.*
>
> **A**: In the supplementary PDF (Sec. 3), we have already provided a detailed comparison of our method against competitive baseline approaches, including parameter counts, computational complexity (measured in GMACs), and inference time. Specifically, our method achieves a processing speed of 34.85 fps on $1280\times720$ video sequences, demonstrating its capability for real-time matting. We also show this table below for convenience.
>
> | Method | #Params (M) | Comp. (GMac) | Infer. (ms) |
> | --- | --- | --- | --- |
> | BGMv2  | 40.25 | 17.47 | 13.27 |
> | MODNet  | 6.49 | 64.95 | 27.90 |
> | RVM  | 26.89 | 11.49 | 8.89 |
> | SGHM  | 40.25 | 25.81 | 25.96 |
> | P3M-Net  | 39.48 | 300.39 | 36.79 |
> | MAM  | 96.45 | 383.02 | 533.2 |
> | SmartMat  | 26.89 | 228.73 | 38.96 |
> | Ours (1st frame) | 70.09 | 60.29 | 34.06 |
> | Ours (other frames) | 70.09 | 44.85 | 28.69 |
>
> We believe this analysis adequately addresses the efficiency of our approach. To ensure this information is not overlooked, we have included the relevant analysis in Sec. 5.9 and presented the comparison results in a new Tab. 5 of the revised manuscript.

---

> ### Author Response · Authors · 2025-08-07
> **Reply to Reviewer WP5u (2/2)**
>
> **RC3**: *The pre-processing pipeline for generating the flash-trimap involves several hyperparameters (e.g., filter r, number of points q). Could the authors investigate the sensitivity of the final performance to these hyperparameters?*
>
> **A**: Thank you for your construction feedback. To fully investigate this, we conducted following ablations on these hyperparameters, specifically the filter kernel size $r$ and the number of points $q$.
>
> |  $r$ | MSE $\downarrow$ | SAD $\downarrow$ | Grad $\downarrow$ | Conn $\downarrow$ |
> |:----------:|:---:|:---:|:----:|:-----:|
> |      3     |  0.652  | 1804.810 |  3741.978 | 1626.490 |
> |      5     |  0.652 | 1804.741 | 3742.230 | 1626.339 |
> |      7     |  0.652  |  1804.740  | 3742.230  | 1626.338 |
> |      9     |  0.652  | 1804.741 |  3742.231 |  1626.339    |
>
> |  $q$ | MSE $\downarrow$ | SAD $\downarrow$ | Grad $\downarrow$ | Conn $\downarrow$ |
> |:-----------:|:---:|:---:|:----:|:-----:|
> |      3      |  1.199 | 2641.574 | 5496.697 | 2682.519 |
> |      4      |  0.686 | 1819.145 | 3797.614 | 1655.620 |
> |      5      |  0.652 | 1804.741 | 3742.230 | 1626.339 |
> |      6      |  0.652 | 1804.742 | 3742.230 | 1626.340 |
>
> Our findings above indicate that using smaller or larger filter kernels has minimal impact on performance. However, larger kernels increase computational cost due to higher sorting complexity, leading us to adopt a median filter kernel size of 5 in this study. Regarding the number of points $q$, smaller values may result in obvious performance degradation, as insufficient confident points may fail to fully represent the foreground in the flash image, leading to incomplete or inaccurate flash cues. Conversely, larger $q$ values have almost no impact on performance. These results have been included in the revised manuscript in Sec. 5.5.2.
>
>
> ### Reference
> [1] Kirillov, A., Mintun, E., Ravi, N., et al.: Segment anything. In: ICCV, pp. 4015–4026 (2023)
> [2] Huang, Y., Yang, H., Sun, K., et al.: Interformer: Real-time interactive image segmentation. In: CVPR, pp. 22301–22311 (2023)

---

### Review · Reviewer_dJWH · 2025-08-21

**Summary Of Contributions:**

The paper proposes using flash/no-flash image pairs for portrait matting instead of clean backgrounds. The authors introduce FNFNet with a transformer-based Foreground Correlation Module using cross-attention and flash ratio maps. The authors create the first flash/no-flash matting dataset (3,025 samples, 133 subjects) and demonstrate superior performance over trimap-free methods on dynamic backgrounds, though underperforming BGMv2 on static scenes.

**Audience:**

Yes

**Claims And Evidence:**

Yes

**Requested Changes:**

- Please provide a compelling analysis of when/why users should choose this method over BGMv2, given the performance degradation on static scenes. Include a decision framework or specific use cases where flash-based matting is superior.
- Please include evaluation on real captured flash/no-flash pairs (not synthetic composites) to validate that the method works in practice. Compare pseudo-flash gamma adjustment against actual flash photography to verify this approximation.
- Please fix the undefined behavior in Eq. 2 when histogram bins approach zero. Add explicit handling of edge cases and empty bins in the flash ratio calculation.
- Please provide more comprehensive failure case analysis beyond the single outdoor example. Characterize limitations systematically (e.g., minimum flash intensity required, maximum pose variation tolerated, temporal constraints).

**Strengths And Weaknesses:**

Strengths
- This paper addresses the real limitation of BGMv2's static background requirement, enabling matting with camera/background movement.
- This paper conducts comprehensive ablations on components, hyperparameters, and multiple baselines with clear metrics.
- The authors introduce the first flash/no-flash matting dataset that fills an important gap for this research direction.
- The FCM's cross-attention mechanism is well-motivated for handling pose variations between flash/no-flash pairs.

Weaknesses
- The proposed method underperforms BGMv2 on static backgrounds without compelling justification for this trade-off.
- Most of the testings are conducted on synthetic composites rather than real flash/no-flash captures. The preprocessing overheads (e.g., SAM) are excluded from the complexity analysis.
- The paper only shows one Single failure case.

---

> ### Author Response · Authors · 2025-09-01
> **Reply to Reviewer dJWH (1/2)**
>
> **W1** & **RC1**: *The proposed method underperforms BGMv2 on static backgrounds without compelling justification for this trade-off. Please provide a compelling analysis of when/why users should choose this method over BGMv2, given the performance degradation on static scenes. Include a decision framework or specific use cases where flash-based matting is superior.*
> **A**: Thank you for your constructive feedback. **Our method’s core advantage lies in its robust handling of dynamic scenes, offering greater flexibility and usability compared to BGMv2, which is limited to static scenes with strict pixel alignment requirements.** While BGMv2 is optimized for static backgrounds, it relies on the critical assumption that background and foreground image pixels are perfectly aligned. When this alignment is disrupted (e.g., due to camera shaking, changes in the background itself, or the appearance of other objects in the background), BGMv2 fails completely. In contrast, our method is designed for matting in dynamic scenes, which can be viewed as a generalization of static scenes. It effectively handles situations that BGMv2 cannot address in static scenarios (such as the examples mentioned). Therefore, in more general and practical applications, our model is inherently more necessary compared to BGMv2. We strongly recommend referring to the video comparison results provided in the supplementary material **(the video file `video_comparison.mp4`)**, where visual comparisons with BGMv2 (particularly in cases involving camera shaking or background changes) better demonstrate the superiority of our method in real-world applications. Additionally, our approach does not impose restrictive requirements such as continuous flash triggering or user calibration. Capturing a flash image is a one-time, automatic process that does not interfere with the user experience. Compared to BGMv2, which requires user intervention prior to each capture, our method offers improved usability and flexibility, making it more suitable for practical deployment.
> We have incorporated the above discussion into Sec. 5.7 and Sec. 5.10 of the revised manuscript.
>
>
> **W2** & **RC2**: *Most of the testings are conducted on synthetic composites rather than real flash/no-flash captures. The preprocessing overheads (e.g., SAM) are excluded from the complexity analysis. Please include evaluation on real captured flash/no-flash pairs (not synthetic composites) to validate that the method works in practice. Compare pseudo-flash gamma adjustment against actual flash photography to verify this approximation.*
> **A**: Thank you for the comment. In fact, we have already provided a comprehensive evaluation of our model on real-world data in the supplementary material PDF. In addition to direct performance comparisons with other baselines on real-world samples, we have already conducted **Environmental Robustness Evaluation** in real-world settings, which fully demonstrates our model's tolerance to specific environmental conditions (e.g., strong lighting environments, backgrounds with specular objects, flash intensity variance, flash color temperature variance, etc.). We also included qualitative and quantitative comparisons of our model in **Real-world Low-light Environments**. In such environments, due to low ambient light, the contrast between foregrounds in the flash image and no-flash image is very strong, which is beneficial for extracting robust guidance for the foreground from the flash image. The experimental results show that our method outperforms baselines in this special environment as well.
> We have also provided video comparison results in the supplementary material (**the video file `video_comparison.mp4`**) to further support the practical applicability of our model in real-world scenarios.
> **These comprehensive evaluations also confirm that our pseudo-flash samples, generated via gamma adjustment, serve as a reliable approximation for actual flash photography.**
> To prevent these experiments from being overlooked, we have incorporated the above experiments from the supplementary material PDF into the revised manuscript.
>
> Furthermore, regarding the preprocessing analysis you mentioned, we have reported the complexity of the entire preprocessing process in Tab. 5 of the revised manuscript. On a PC accelerated by a single A5000 GPU, the entire preprocessing requires only 0.12 seconds. However, it should be noted that in scenarios with strict real-time requirements, such as video conferencing or live-streaming, the flash cue extracted from a single flash image during our preprocessing can guide matting for multiple no-flash images in the same scene. Therefore, the relatively longer preprocessing time does not impact our model in mainstream applications, ensuring that our method maintains high efficiency without compromising performance. We have added the above discussion to Sec. 5.9 of the revised manuscript.

---

> > ### Author Response · Authors · 2025-09-01
> > **Reply to Reviewer dJWH (2/2)**
> >
> > **RC3**: *Please fix the undefined behavior in Eq. 2 when histogram bins approach zero. Add explicit handling of edge cases and empty bins in the flash ratio calculation.*
> > **A**: Thank you for highlighting this. Indeed, the denominator in Eq. 2 may approach zero, leading to cases where the flash ratio $r$ could become $-\infty$. However, this edge case is addressed by Eq. 3, which corrects the flash ratio calculation. Since the truncation threshold $\delta$ in Eq. 3 is positive, any $-\infty$ values are truncated to zero, effectively resolving these edge cases. To clarify this, we have added a detailed discussion to Sec. 4.2.2 of the revised manuscript.
> >
> > **W3** & **RC4**: *The paper only shows one Single failure case. Please provide more comprehensive failure case analysis beyond the single outdoor example. Characterize limitations systematically (e.g., minimum flash intensity required, maximum pose variation tolerated, temporal constraints).*
> > **A**: Thank you for your constructive feedback. Here we provide more comprehensive extreme cases analysis regarding both tolerance and failure in extreme scenarios of our model.
> >
> > - **Model Tolerance to Challenging Factors:** Our model demonstrates robust tolerance to the challenging factors highlighted, including flash intensity variation, pose variation, and temporal constraints. To validate this, we conducted extensive experiments detailed in the revised manuscript. For flash intensity, results are presented in Sec. 5.6 and Fig. 11 of the revised manuscript, where we demonstrate our model’s performance under different lighting environments and flash conditions. Specifically, Fig. 11(a) shows robustness of our model to low flash contrast. And for pose variation and temporal constraints, the video comparisons in the supplementary material (**the video file `video_comparison.mp4`**) showcase our model’s effectiveness in real-world dynamic scenes, including scenarios with foreground object motion. Despite processing video data on a per-frame basis without explicit temporal modeling, the results demonstrate strong temporal consistency, confirming the model’s practical applicability in dynamic settings.
> >
> > - **Extreme Cases Analysis:** Our model performs reliably across a wide range of typical scenarios; however, failures may occur under specific, extreme conditions, which we systematically categorize into two aspects: (1) **violations of the flash photography assumption** and (2) **challenges with extreme temporal conditions**.
> >   - **Violations of Flash Photography Assumption**: The flash photography assumption may violate in some rare scenarios where the flash/no-flash contrast is severely disrupted. These conditions lead to inaccurate flash ratio map estimations, resulting in unreliable matting outputs. Such failures occur only in these extreme cases, as our method otherwise robustly leverages flash-induced contrast for foreground extraction.
> >   - **Challenges with Extreme Temporal Conditions**: Other extreme cases, such as large motion variation or blurred images caused by extreme temporal conditions (e.g., fast movement), can also cause our model to produce suboptimal results. This limitation arises because our method processes frames independently, without explicit temporal modeling, making it sensitive to significant motion or image degradation in such rare scenarios.
> >
> >   To illustrate these edge cases, we have added two cases in Fig. 15 to demonstrate the impact of large motion variation and motion blur on our model respectively. To address them in future work, we plan to integrate motion cues from flash/no-flash pairs and explore techniques such as a memory bank to explicitly model temporal information, further enhancing robustness for extreme pose variations while maintaining the model’s strong performance in typical scenarios.
> >
> > These discussions and results have been incorporated into Sec. 5.10 and Sec. 6 of the revised manuscript.

---

### Decision · Action_Editor_9rdz · 2025-11-28

**Recommendation:** Accept as is

**Additional Comments:**

The authors have addressed the concerns raised by the reviewers well in their rebuttal. The work can be published as is.

**Audience:**

Yes

**Audience Explanation:**

The authors propose an approach that relies on flash/no-flash image pairs instead of clean/static backgrounds for image matting. This is both interesting and clever. One of the reviewers points out that the scope is limited to portrait matting with specialized hardware instead of more general computer vision. Nevertheless, the reviewers found that the paper has merits, and the methodology should be of interest to the TMLR community.

**Claims And Evidence:**

Yes

**Claims Explanation:**

Claims in the paper are well-supported by comprehensive validation. As the reviewers also point out, the methodology and data are presented in good detail, and the validation/experiments are extensive.